# Deep Learning-Based Change Detection in Remote Sensing Images: A Review

**Ayesha Shafique [1], Guo Cao [1,*], Zia Khan [2], Muhammad Asad [3] and Muhammad Aslam [4]**

1   School of Computer Science and Engineering, Nanjing University of Science and Technology, Nanjing 210094, China; ayeshashafique@njust.edu.cn
2   Department of Computer Science, Central South University, Changsha 410083, China; zia.official@csu.edu.cn
3   Department of Computer Science, Nagoya Institute of Technology, Nagoya 466-8555, Aichi, Japan; m.asad@itolab.nitech.ac.jp
4   School of Computing, Engineering and Physical Sciences, University of West of Scotland, Blantyre, Glasgow G72 0LH, UK; muhammad.aslam@uws.ac.uk
*   Correspondence: caoguo@njust.edu.cn

**Abstract:** Images gathered from different satellites are vastly available these days due to the fast development of remote sensing (RS) technology. These images significantly enhance the data sources of change detection (CD). CD is a technique of recognizing the dissimilarities in the images acquired at distinct intervals and are used for numerous applications, such as urban area development, disaster management, land cover object identification, etc. In recent years, deep learning (DL) techniques have been used tremendously in change detection processes, where it has achieved great success because of their practical applications. Some researchers have even claimed that DL approaches outperform traditional approaches and enhance change detection accuracy. Therefore, this review focuses on deep learning techniques, such as supervised, unsupervised, and semi-supervised for different change detection datasets, such as SAR, multispectral, hyperspectral, VHR, and heterogeneous images, and their advantages and disadvantages will be highlighted. In the end, some significant challenges are discussed to understand the context of improvements in change detection datasets and deep learning models. Overall, this review will be beneficial for the future development of CD methods.

**Keywords:** change detection methods; remote sensing images; SAR image; multispectral images; hyperspectral images; VHR images; heterogeneous image; deep learning

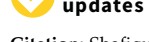



## 1. Introduction and Background

With the advancement of Remote sensing(RS) technology, RS platforms have become increasingly capable of collecting a wide range of data. These available data have become key resources for environmental monitoring by detecting changes on the land surface. Change detection (CD) is a phenomenon of detecting change of the same geographical area by observing the set of images captured at different periods [1–3]. It has attracted widespread interest due to it being extensively used in several real-world applications, such as fire detection, environmental monitoring [4], disaster monitoring [5], urban change analysis [6], and land management [7], among others. Therefore, CD has attracted increasing attention from researchers throughout the world.

RS data contain limited temporal, spatial, and spectral resolutions that significantly constrain RS-based CD methodologies. However, the development of sensors with greater technical capabilities has overcome many of these constraints. As a result, researchers have examined an ever-expanding set of methodologies, algorithms, and procedures for detecting change. In RS, numerous types of satellites have been launched in space, such as active or passive, optical, or microwave sensors, and have high- or low-resolution. The satellite datasets are valuable data sources for describing urban land use/cover types and their changes over time. Various approaches for detecting changes in satellite images

have been developed to find changes in the status of an object or phenomenon [8]. For CD, multimodal RS images, such as synthetic aperture radar (SAR) [9,10], multispectral (MS), or hyperspectral (HS) images have been used, which are acquired from an active sensor (SAR), passive optical sensor (MS), and others.

Over the last several decades, various CD approaches have been developed. Traditional CD approaches may be classified into two groups based on the analysis unit: pixel-based CD (PBCD) and object-based CD (OBCD). PBCD is the conventional approach and identifies changes by comparing pixels, and as a result, it cannot overcome the limits of radiometric variations and misregistration across various dates or sensors. Because of the increased variability among image objects, PBCD approaches, which are generally appropriate for middle- and low-resolution RS images, frequently fail to operate in VHR imagery. The OBCD solves these issues and increases CD accuracy significantly. OBCD-based techniques are proposed for VHR image CDs, where images are segmented into disjoint and homogeneous objects, then bitemporal objects are compared and analyzed. Different approaches of PBCD and OBCD are shown in Figure 1.

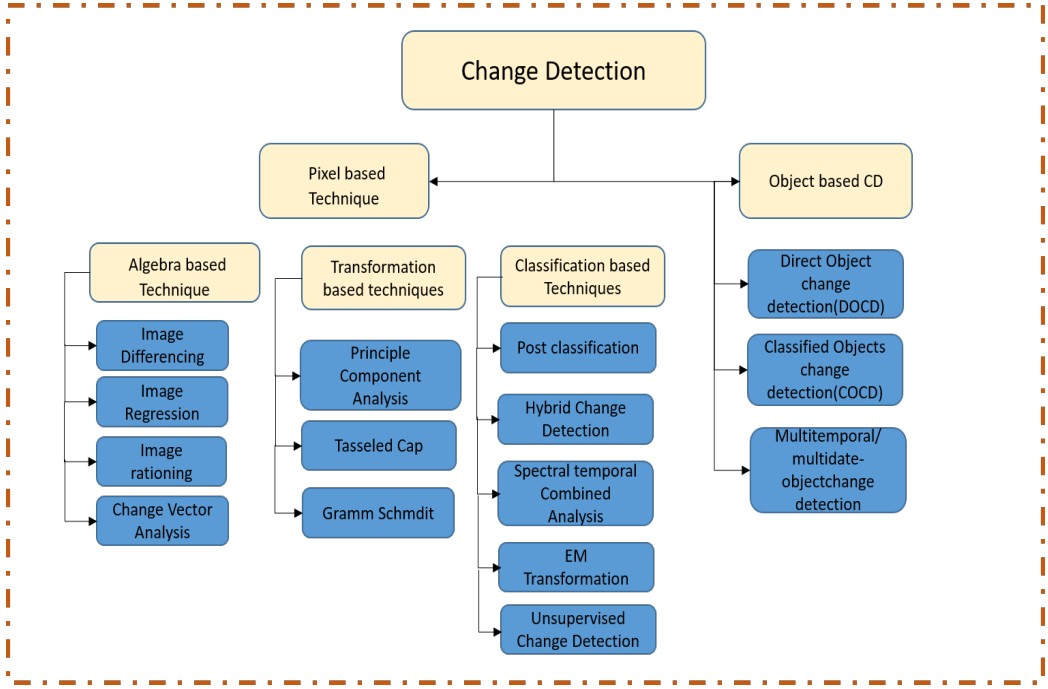

**Figure 1.** Traditional change detection method.

Many traditional techniques of change detection, such as image algebra [11] and transformation [12], are relatively limited in their applications and are affected by the influences of atmospheric conditions, changing seasons, satellite sensors, and solar elevations, reducing the accuracy of change detection [13,14]. While specific approaches, such as object-based image analysis, reduce false changes by extracting geometric and textural characteristics, this needs a tiresome and time-consuming procedure. They effectively eliminate the benefit of an automated CD approach. Furthermore, identifying change regions is difficult due to the threshold selection [15]. The unsupervised method is presented to facilitate change detection by using the expectation–maximization algorithm for threshold selection. However, selecting suitable criteria to capture all change regions while eliminating undesired ones remains challenging. In addition, the classification-based CD approaches [16] convert pixels of an image into land cover classes without the need for random noises for change identification [17–19]. Most previous supervised or unsupervised techniques rely on hand-crafted feature representations, which have limited representational abilities to describe complicated and high-level change information, resulting in low performance under clutter land covers. All of the previously described classification methods would be

appropriate for training samples. However, those methods cannot incorporate specific and dependable statistical features for many datasets and, hence, do not produce high detection performance for new datasets. Each scheme focuses on only a few aspects of the domain while ignoring others. In the absence of a clear and explicit definition that integrates all relevant aspects, the concept of an RS CD technique remains fuzzy.

Deep learning methods automatically discover the representations of input data required for the CD. Recently, a DL-based CD in the RS sector has become a "hotspot", attracting major attention and yielding good results. In recent years, to identify changes in RS images, DL can automatically derive complicated, hierarchical, and non-linear features from raw data and overcome several limitations of traditional change detection methods. Because of their tremendous modeling and learning capabilities, deep learning approaches represent the link between the image object and its real-world geographical elements as closely as possible, allowing for more real-world change information [20,21].

### 1.1. Contribution of This Study

Various survey studies have been published in the literature over the last decade, reviewing various machine and deep learning models for change detection and almost all published reviews in the area of remote sensing image CDs were considered to show the overall picture of research contributions in the field. Table 1 summarizes the works that have been published thus far. One paper [22] published in reputable journals has focused on DL-based methods for remote sensing. Review papers [23] discussed pixel-based to object-based change detection methods and highlight some issues. In [24], the authors reviewed the AI-based CD technique. The authors of [25] reviewed the analysis on CD techniques for RS applications, but they did not highlight the challenges of the current CD method in deep learning. Some review papers [26] have focused on hyperspectral and multispectral image CDs, but still, there are a lack of reviews in which all RS datasets are discussed for change detection in deep learning, highlighting its challenges.

**Table 1.** Summary of survey papers on change detection.

| Reference | Publisher | Publication Year | Citation |
| --- | --- | --- | --- |
| [2] | Taylor & Francis | 1989 | 4834 |
| [27] | Taylor & Francis | 2002 | 60 |
| [28] | Taylor & Francis | 2004 | 3595 |
| [29] | Taylor & Francis | 2012 | 510 |
| [23] | Elsevier | 2013 | 1219 |
| [30] | Elsevier | 2016 | 134 |
| [31] | IEEE | 2016 | 1295 |
| [22] | others | 2017 | 411 |
| [20] | IEEE | 2017 | 1475 |
| [32] | Elsevier | 2018 | 124 |
| [33] | IEEE | 2019 | 90 |
| [21] | ISPRS | 2019 | 689 |
| [26] | MDPI | 2019 | 11 |
| [34] | IEEE | 2020 | 48 |
| [24] | MDPI | 2020 | 76 |
| [25] | Elsevier | 2020 | 8 |

To the best of our knowledge, no work has explored the current advancements and mostly-used RS datasets, such as SAR, multispectral, hyperspectral, VHR, and heteroge-

neous images for CD by using deep learning separately, and presenting it in its categories, such as supervised, unsupervised, and semi-supervised, in a particular and comprehensive manner at one platform. None of the reviews have discussed CD from the perspective of datasets separately based on the deep learning method. We present the data from different sensors used for CD in detail, mainly including SAR, multispectral, hyperspectral, VHR, and heterogeneous images. We also highlight some challenges at the end of this review that need to be solved. This review will be helpful to new researchers regarding the issues and challenges faced by the community in this domain. Moreover, this paper analyzes the research gaps found in the literature that will help future researchers identify and explore new avenues in the area of RS CD for deep learning.

### 1.2. Organization of This Work

The outline of this literature review is as follows. Section 2 explains the inclusion and exclusion criteria. Section 3 highlights the remote sensing sensors and dataset analysis, covering the datasets of SAR, multispectral, hyperspectral, VHR, and heterogeneous images. Section 4 covers the change detection architecture. Section 5 covers the literature review of the change detection in remote sensing datasets by using DL-based networks. Section 6 expresses the evaluation techniques. Section 7 highlights the discussion and, finally, Section 8 concludes the paper.

## 2. Research Inclusion/Exclusion Criteria

Firstly, search criteria were set to extract the maximum publications from the selected sources. The selection criteria were divided into three parts in order to collect more relevant articles. The title of the article was verified in the first phase to eliminate duplicate and irrelevant papers. We studied the abstracts of the articles collected in the first step, in the second stage, to pick relevant papers to the specified field. At the end of the process, we examined each paper in depth and finalised the papers for this study. We reviewed the significant research papers in the field published during 2015–2021, mainly from the years of 2019 and 2020, with some papers from 2021. The main focus was papers from the most reputed publishers, such as Elsevier, IEEE, and Remote Sensing. Some of the latest papers were selected from conferences, such as IGRASS and ISPRS. We reviewed 68 papers on deep learning change detection for each type of dataset. Our focus was to present a review on RS datasets for CD in deep learning. Figure 2 presents the details of the year-wise publications. Moreover, datasets and category wise discussions are presented in Sections 5.1–5.5.

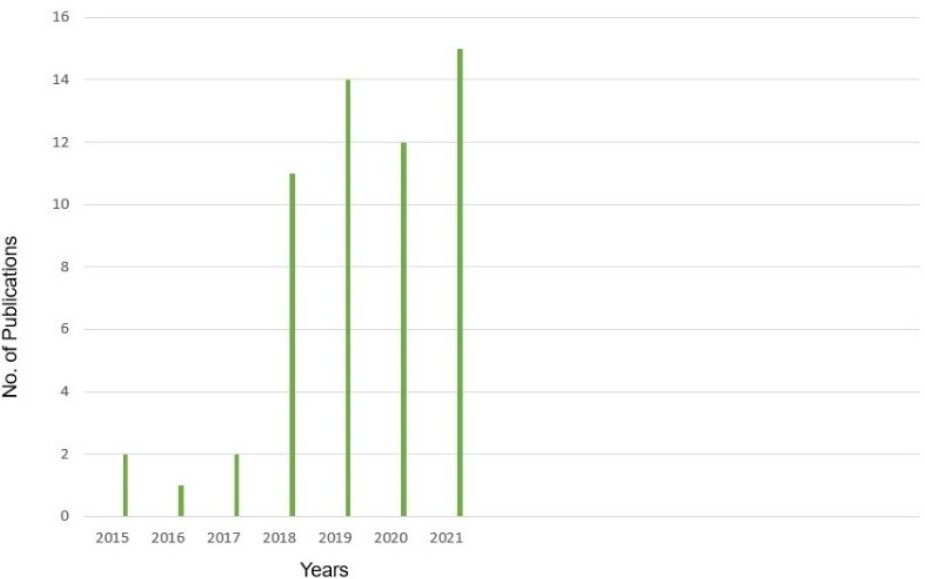

**Figure 2.** Year-wise publications in journals from 2015 to 2021.

### 3. Remote Sensing Datasets for Change Detection

*3.1. Sensors for Collecting Change Detection Datasets*

Different kinds of satellites have recently been launched into space, including active and passive, optical and microwave sensors, and high- and low-resolution. Satellite images are good sources of data for determining urban land use/cover forms and how they vary over time and space [35]. Different RS devices are installed on a satellite to collect data about an object on Earth's surface without direct physical contact with the object. When compared to aerial and terrestrial platforms, spaceborne platforms are the most stable carriers. There are two modes of interaction between a sensor and the Earth's surface: active and passive. Active sensors produce energy to illuminate objects and measure observations. Passive satellite sensors include the Landsat, GeoEye, SPOT, EROS, and WorldView spacecrafts [36]. Due to recent sensor developments, such as TerraSAR-X [37] and COSMO-SkyMed, which operate at sub-metric resolutions, or very high temporal resolutions, such as Sentinel-1, and a series of systems that ensure continuous measurements over several decades, we can now consider operational applications in many civilian sectors, e.g., M environmental surveillance and the security of goods and people, as well as many other geophysical sciences fields. A SAR image's key properties are its resolution and geographic coverage, as well as the frequency range, incidence angle, and emission/reception polarisation of the obtained signal. In general, the use of a SAR image in RS necessitates geometric and radiometric calibrations, which can be accomplished in accordance with the methods described by the agencies sending the data coverage. It is necessary to conduct geometric and radiometric calibrations on SAR images before using them in remote sensing.

However, MS RS is advantageous in terms of data availability. MS RS systems use parallel frame sensors to detect radiation in a few bands, usually three to six spectral bands from visible to middle infrared. Aside from these bands, various kinds of satellite sensors capture images in one or two thermal bands. As a result, MS satellite sensors have fewer but broader spectral bands, which cannot detect tiny details on the land's surface and do not allow the separation of objects with minor spectral reflecting differences [38]. While hyperspectral remote sensing sensors can capture images in a variety of narrow spectral bands found in the electromagnetic spectrum, ranging from visible to near-infrared, medium infrared, and thermal infrared, HS sensors can capture energy in 200 bands or more, implying that they continuously cover the reflecting spectrum for each pixel in the scene. There are two distinct types of images capture systems used in hyperspectral imaging: aircraft (AVIRIS) and satellite-based (Hyperion on EO-10). The majority of hyperspectral sensors are deployed on aerial platforms, with fewer on satellites [39–42].

As earth observation technology advances, more advanced satellite sensors, such as QuickBird, SPOT, and others, are becoming available, and are designed to gather VHR images. Table 2 summarizes the most regularly utilized RS satellites and their specifications.

**Table 2.** Provides details of the most widely utilized RS satellites and their characteristics [43–45].

| Satellite/Sensors | Country | Year | Revisit (Day) | Spatial Resolution (m) |
|---|---|---|---|---|
| COSMO | Italy | 2010 | 5 | 15 m |
| Gaofen 3 | China | 2016 | 5 | 1–500 m |
| Landsat 9 | USA | 2020 | 16 | 15 m |
| TerraSAR-X | Germany | 2007 | 2.5–11 days | 1–16 |
| SPOT7 | USA | 2014 | 1–3 | 1.5 m |
| ERS2 | ESA | 1995 | 336 | 6–30 m |
| RADARSAT | Canada | 2018 | 1 | 3–100 m |
| Hyperion (EO-1) | USA | 2000 | 2–16 | 30 m |

**Table 2.** *Cont.*

| Satellite/Sensors | Country | Year | Revisit (Day) | Spatial Resolution (m) |
|---|---|---|---|---|
| ALOS | Japan | 2006 | 2 days | 2.5, 10 m |
| IKONOS | USA | 1999 | 3 | 1 m, 4 m |
| QuickBird | USA | 2001 | 2.4–5.9 | 2.61 m |
| Envisat | ESA | 2002 | 35 days | 300 m |
| GeoEye | USA | 2008 | 8.3 | 0.41 m |
| WorldView 1 | USA | 2007 | 1.7 | 0.5 m |
| WorldView 2 | USA | 2009 | 1.1 | 0.46 m |
| WorldView 3 | USA | 2014 | <1 | 1.24 m |
| WorldView 4 | USA | 2016 | 3 | 0.34 m |
| Sentinel-1 | ESA | 2014 | 12 | 5–20 m |
| Sentinel-2 | ESA | 2015 | 10 | 10–60 m |
| Sentinel-3 | ESA | 2016 | 27 | 5–40 m |
| Sentinel-4 | ESA | 2019 | 1 | 10 m |
| Sentinel-5 | ESA | 2014 | <1 | 20 m |
| Sentinel-6 | ESA | 2020 | 9 | 60 m |

*3.2. Datasets for Change Detection*

This section explains different types of commonly used RS datasets for CD, such as SAR, multispectral, hyperspectral, VHR, and heterogeneous images.

### 3.2.1. SAR Images

SAR is a type of radar that produces a two-dimensional image or three-dimensional reconstructions of an object, such as landscapes [46]. It is a technique for remotely mapping the reflectivity of objects or their surroundings with high spatial resolutions, by sending and receiving electromagnetic (EM) signals. The images obtained using this method can be used for a wide range of applications, from basic radar functionalities, such as object detection and geographical localization, to estimating some geophysical properties of complex environments, such as certain dimensions, roughness, moisture content, density, etc. SAR data are captured from an active microwave sensor that reflects backscattering information of land cover in all weather conditions and at all times. As a result of its independence from sunshine conditions, the SAR imagery provides an advantage for change detection tasks. Due to the frequency domains commonly employed by SAR devices, this active sensing technology is not affected by sunlight but only slightly by the weather. SAR systems based on aerial or satellite sensors quickly scan huge regions impossible to reach using field measurements. This type of characteristic makes radar imaging systems a particularly well-suited instrument for RS [47]. Moreover, the negative effects of geometric distortion and electromagnetic interference, such as target overlap, perspective shrinkage, and multipath effect, must be addressed in the SAR [48]. Table 3 lists the most often used CD datasets for SAR, including the Ottawa dataset, Bern dataset, [49], San Francisco datasets [50], Farmland, and others. The links of the datasets are as follows:

1.  Bern dataset open source: https://github.com/yolalala/RS-source (accessed on 22 December 2021).
2.  San Francisco dataset open source: https://github.com/yolalala/RS-source (accessed on 22 December 2021).
3.  Farmland dataset open source: https://share.weiyun.com/5M2gyVd (accessed on 22 December 2021).

**Table 3.** Illustration of SAR datasets.

| Satellite | Area | Dataset Name | Image | Pixel | Date |
|---|---|---|---|---|---|
| RADARSAT SAR | Canada | Ottawa |  | 290 × 350 | May–August 1997 |
| RADARSAT-2 | China | Yellow river |  | 257 × 289 | June 2008 June 2009 |
| ERS-2 | US | San Francisco |  | 256 × 256 | August 2003 May 2004 |
| Landsat ETM+ | Mexico | Mexico |  | 512 × 512 | April 2000 March 2002 |
| ERS-2 | Switzerland | Bern |  | 301 × 301 | April–May 1999 |
| Envisat | Japan | Sulzberger |  | 256 × 256 | March 2011 |
| RADARSAT-2 | China | Beijing |  | 1024 × 1024 | October 2010 |

Multi-band, multi-polarization, multi-platform SAR images are becoming increasingly common as SAR imaging technology progresses, providing more data sources for CD tasks. On the other hand, SAR images always have speckle noise, making change detection more difficult than optical RS images.

### 3.2.2. Multi-Spectral Images

A multispectral image (MSI) gathers image data across the electromagnetic spectrum at certain wavelength ranges. MSI can extract additional information that the human eye fails to capture with its visual receptors for red, green, and blue [51]. The wavelengths may be separated by filters or detected using sensitive instruments, including light from frequencies beyond the visible light range, i.e., infrared and ultraviolet. MSI is often acquired using a passive optical sensor that gathers information about ground objects in many spectral bands. MSI with spatial resolutions ranging from low to high may be obtained cheaply and consistently. Vibrant texture, colors, and other features are also provided. MSI measures the radiation inherent to an object, regardless of an external light source [52–55]. It includes acquiring visible, near-infrared, and short-wave infrared images in a few broad wavelength bands. Different materials reflect and absorb differently at different wavelengths. The resolutions of remote sensing images vary; the application scenarios for the various resolution multispectral images differ slightly. MS images are generally used for CD, and the most often used MS images for deep learning-based CD algorithms are taken from Landsat [56–63] and the Sentinel series [64,65] of satellites because their low collections cost great temporal and spatial coverage. Furthermore, additional satellites, such as QuickBird, SPOT [66–68], Gaofen [69,70], and Worldview, provide very high spatial resolution images, while others provide very high spatial resolution aerial [71] images, allowing the CD findings to preserve more details of the changes. The multispectral datasets are classified into wide and local area change datasets.

### 3.2.3. Wide-Area Datasets

It concentrates on changes throughout a large coverage region while neglecting the details of sporadic targets. Multispectral (MS) datasets are obtained by satellites carrying the imaging spectrometer as the most accessible and intuitive remote sensing images. The EROS Data Center's Southwest U. S. CD Images [72] are the first open-source dataset for the CD task that uses a change vector to explain changes in greening and brightness.

The datasets that fall under it are Southwest U. S. Change Detection Dataset, MtS-WH, Taizhou dataset [73], Onera Satellite Change Detection dataset [74], and NASA Earth Observatory Change Datasets. Some examples of wide area datasets are mentioned in Table 4.

**Table 4.** Illustration of wide area datasets.

| Satellite | Area | Dataset Name | Image | Pixel | Date |
|---|---|---|---|---|---|
| World View-2 | Los Angeles | U. S. | | 322 × 266 | 1986–1992 |
| Landsat 7 | China | Kunshan | | 400 × 400 | March 2000 February 2003 |
| IKONOS | China | MtS-WH | | 7200 × 6000 | February 2002 June 2009 |
| Sentinel-2 | UAE | Onera | | 700 × 700 1200 × 1200 | 2015–2018 |

The links of Wide-area datasets are as follows

1. Southwest U. S. dataset open source: https://geochange.er.usgs.gov/sw/changes/anthropogenic/vegas (accessed on 22 December 2021).
2. MtS-WH dataset open source: Open source: http://sigma.whu.edu.cn/newspage.php?q=2019-03-26 (accessed on 22 December 2021).
3. NASA Earth Observatory dataset open source: https://earthobservatory.nasa.gov/images/146194/how-cancun-grew-into-a-major-resort (accessed on 22 December 2021).
4. Onera Satellite dataset open source: https://ieee-dataport.org/open-access/oscd-onera-~satellite-change-detection (accessed on 22 December 2021).

### 3.2.4. Local Area Datasets

It is vital to investigate certain goals in an urban region, such as buildings, rivers, roads, etc. HR RS images are the primary data sources for artificial neural network changes of target and detail regions. The datasets that fall under it are: SZTAKI dataset, Hongqi Canal dataset, Minfeng dataset, season changes detection dataset, HRSCD dataset, and building CD dataset [75–78]. Regardless of the application scenario, MSI has inherent restrictions. The accuracy of the feature extraction method will be affected not only by weather circumstances, such as mist and fog, but also by the difference in shooting time. The results of the changes are affected by shadows and distractions near the concerned targets. Some examples of local area datasets are mentioned in Table 5.

**Table 5.** Illustration of local area datasets.

| Satellite | Area | Dataset Name | Image | Pixel | Date |
|---|---|---|---|---|---|
| FÖMI | China | SZTAKI | | 952 × 640 | 2000 2003 |
| Géoportail | China | HRSCD | | 321 × 330 | 2002 2005 |
| WorldView-2 | China | Yandu | | 322 × 350 | 19 September 2012 10 February 2015 |

The links of local area datasets are as follows:

1. HRSCD dataset open source: https://ieee-dataport.org/open-access/hrscd-high-resolution-semantic-change-detection-dataset.

2. SZTAKI dataset open source: http://web.eee.sztaki.hu/remotesensing/airchange~benchmark.htm (accessed on 22 December 2021).
3. Season changes dataset open source: https://drive.google.com/file/d/1GX656JqqOyBi-Ef0w65kDGVto-nHrNs9 (accessed on 22 December 2021).
4. Building change dataset open source: https://study.Rsgis.whu.edu.cn/pages/download/building-dataset.html (accessed on 22 December 2021).

### 3.2.5. Hyperspectral Images

Hyperspectral images (HSIs) are captured using HSI sensors. HSI technology aims to capture hundreds of spectral channels from the Earth's immediate surface that can precisely characterize the chemical composition of various materials. The spatial and spectral resolutions of HSI distinguishes them. The geometric relationships of image pixels (to one another) are determined by spatial resolution, whereas spectral resolution determines changes within image pixels as a function of wavelength.

Most HS optical passive sensors measure the reflectance of an object in the visible (0.4–0.7 μm) to short-wave infrared (IR) spectrum (2.5 μm). The sensor samples the reflected radiance with excellent spectral resolution (e.g., 10 nm). This dense spectrum sampling enables an accurate representation of each pixel's reflectance, resulting in a precise measurement of the spectral signature [33]. Because multispectral sensors provide coarse spectral sampling in a few distinct spectrum bands, the early stage of developing CD algorithms for these images focuses largely on identifying strong, abrupt, and sudden changes. The purpose of CD in HS is to detect changes associated with significant spectral fluctuations and those associated with minor spectral variations using comprehensive spectral sampling of HS sensors (which are usually not detectable in MS images). Most of the time, these changes only influence a subset of the spectral signatures. In Table 6, some examples of hyperspectral datasets are mentioned.

**Table 6.** Illustration of hyperspectral datasets.

| Satellite | Area | Dataset Name | Image | Pixel | Date |
|---|---|---|---|---|---|
| Hyperion sensor | China | Jiangsu Province |  | 420 × 140 | 3 May 2006 |
| Hyperion sensor | USA | Hermiston City |  | 308 × 350 | 23 April 2007<br>1 May 2004<br>8 May 2007 |
| AVIRIS | Oregon | Hermiston dataset |  | 390 × 200 | 2007–2015 |
| AVIRIS | California | Santa Barbara dataset |  | 984 × 740 | 2013–2014 |
| AVIRIS | USA | California |  | 147 × 316 | 21 October 2015<br>25 June 2018 |
| EO-1 | Henan | Dalin |  | 187 × 268 | 6 March 2003<br>16 April 2006 |

HSI can provide more spectral information than SAR [33] and MS images [79]. As a result, HSIs may detect finer variations [80], reflecting the exact composition of distinct objects. Though HSI-CD approaches have been used, this does not indicate that the hyperspectral CD difficulties may be resolved, as change detection is a challenging process that various factors can influence. The primary challenges for HSI-CD are summarised below.

### 3.2.6. Limited Labeled Data

Because deep learning models have millions of parameters, it is not easy to train them without labeled data. Furthermore, categorizing each pixel in the HSI dataset is time-consuming and requires human experts [81].

### 3.2.7. High-Dimensionality

Due to high-dimensionality of the hyperspectral dataset, some CD algorithms are difficult to handle. Even though band selection and feature extraction are used to minimize dimensionality, some precise details information can also be lost to some extent. The development of different sensors and platforms creates different challenges, e.g., high-dimensional datasets (hyperspectral features and high spatial resolution), complex data structures, and high computational complexity. The high-dimensional remote sensing data HSI have limited accessibility to training samples, which makes deep neural networks, as a result, generally unsuccessful at generalizing the partitioning of HIS data during the training stage [82].

### 3.2.8. Mixed Pixels Problem

Mixed pixels problems typically exist in real HSI-CD, and these pixels consist of several distinct substances. It is not easy (and sometimes impossible) to characterize a mixed pixel accurately if it is roughly classified as a specific type of substance [83].

### 3.3. Very High Spatial Resolution (VHR) Images

VHR images from several satellite sensors are becoming more commonly available, substantially improving the data sources for CD. VHR is collected by many satellite sensors, including QuickBird, GaoFen, and others. VHR images can provide more information about surface characteristics and spatial distributions than medium- and low-resolution images. CD in remotely sensed VHR images has an enormous effect in building CD research, urban expansion, and urban internal change analysis; hence, it has attracted the interest of many researchers. In Table 7, some examples of VHR datasets are mentioned

**Table 7.** Illustration of very high spatial resolution datasets.

| Satellite | Area | Dataset Name | Image | Pixel | Date |
|-----------|------|--------------|-------|-------|------|
| Worldview-2 | Italy | VHR World View 2 | | 420 × 140 | August 2010 May 2011 |
| QuickBird | Italy | image pair | | 1400 × 1400 | August 2012 September 2013 |
| Google Earth | US | LEVIR-CD dataset | | 1024 × 1024 | 2002–2018 |

Although employing VHR to identify change is favorable, there is a technological challenge due to the following.

### 3.3.1. Limited Spectral Information

Images acquired by VHR sensors have fewer bands than images captured by medium-resolution sensors. VHR sensor WorldView-3 can provide images with up to 16 spectral bands; other VHR images, such as QuickBird, IKONOS, Ziyuan-3, and WorldView-2, only cover four bands [84]. It is hard to distinguish classes with equivalent spectral signatures with limited spectral information because of the low between-class variation. Researchers have also stated that high-accuracy change detection is difficult due to the limited spectral information of VHR images [18,85–88]. The direct use of traditional spectral-based CD approaches, such as change vector analysis, is difficult. Consequently, different features are frequently used to enrich spectral information for VHR CD.

### 3.3.2. Spectral Variability

The spectral variability of VHR images is high. Buildings, for example, have complex appearances due to various roof superstructures, such as pipelines, chimneys, and water tanks; as a result, spectral characteristics in VHR images are significantly heterogeneous [89,90]. High spectral variability within geographic objects increases within-class variance, resulting in the uncertainty of spectral-based image interpretation methods. Weather, sun angles, phenological stages, tidal stages, soil moisture, and water turbidity (atmosphere and external factor) can cause unchanging objects to have temporally varying spectral properties, misclassifying them as changed [91,92]. Furthermore, temporary objects visible in VHR imagery, such as cars on the road, can influence the efficacy of spectral-based CD in VHR images.

### 3.3.3. Information Loss

The existence of cloud shadow/haze, terrain, tree shadows, building, and VHR images suffer from major information loss. By selecting cloud-free observations, the problem of clouds and their shadow contamination can be avoided [93]. When tall objects in the image block sunlight, shadows arise, resulting in a partial or even complete loss of information from the Earth's surface covered by shadows. Shadows cast by terrain, buildings, and trees, on the other hand, appear inevitable in VHR imagery, particularly in metropolitan environments [94]. Though shadow information is significant in building detection and height estimation [95,96], it becomes an issue in larger areas for change detection [97].

### 3.3.4. Heterogeneous Datasets

The heterogeneous images are captured by different sensors, resolutions, frequency (SAR) illumination conditions, and polarization. At the moment, it is a more convenient and adaptable method of obtaining heterogeneous images from multiple sensors to obtain multi-temporal images with a higher shooting frequency. However, the CD of the heterogeneous image is not being promoted due to the technological difficulties in data processing of multiple sensor data. The relevant dataset has not yet been adequate. The different feature representations of ground objects in the image obtained by multiple sensors, particularly those captured from optical and SAR sensors, make heterogeneous CD difficult [98,99]. However, calculating the difference between heterogeneous images is difficult because direct comparisons are impossible. After all, heterogeneous images represent unique physical qualities of the objects and display quite diverse statistical tendencies. In Table 8, some commonly used heterogeneous images are elaborated.

**Table 8.** Illustration of Heterogeneous datasets.

| Satellite | Area | Dataset Name | Image | Pixel | Date |
|-----------|------|--------------|-------|-------|------|
| ETM+ | US | Mexico | | $512 \times 512$ | April 2000 March 2002 |
| LANDSAT 7 | China | Farmland | | $306 \times 291$ | 2008 2009 |
| LANDSAT 7 | China | Shuguang Village | | $921 \times 593$ | June 2008 September 2012 |
| Gaofen-3 | China | Sichuan Province | | $2827 \times 1333$ | 24 June 2017 |
| Landsat-5 | Italy | Sardinia | | $412 \times 350{,}300$ | September 1995 July 1996 |

## 4. Change Detection Architecture

With the advent of civilian remote sensing, the world has benefited from constant and expanding surveillance through satellite imagery. This coverage is accomplished using various sensors with varying time, spatial, and spectral scale properties. These properties make it possible to characterize a broader range of earth surface elements and alter processes. Change identification was also constrained by the availability and accuracy of data. Indeed, remote sensing data have a broad range of applications in CD, which requires detecting major changes in output images. Thus, Figure 3 demonstrates the general flow of change detection based on deep learning, and this section explains the components of the change detection architecture and their interactions.

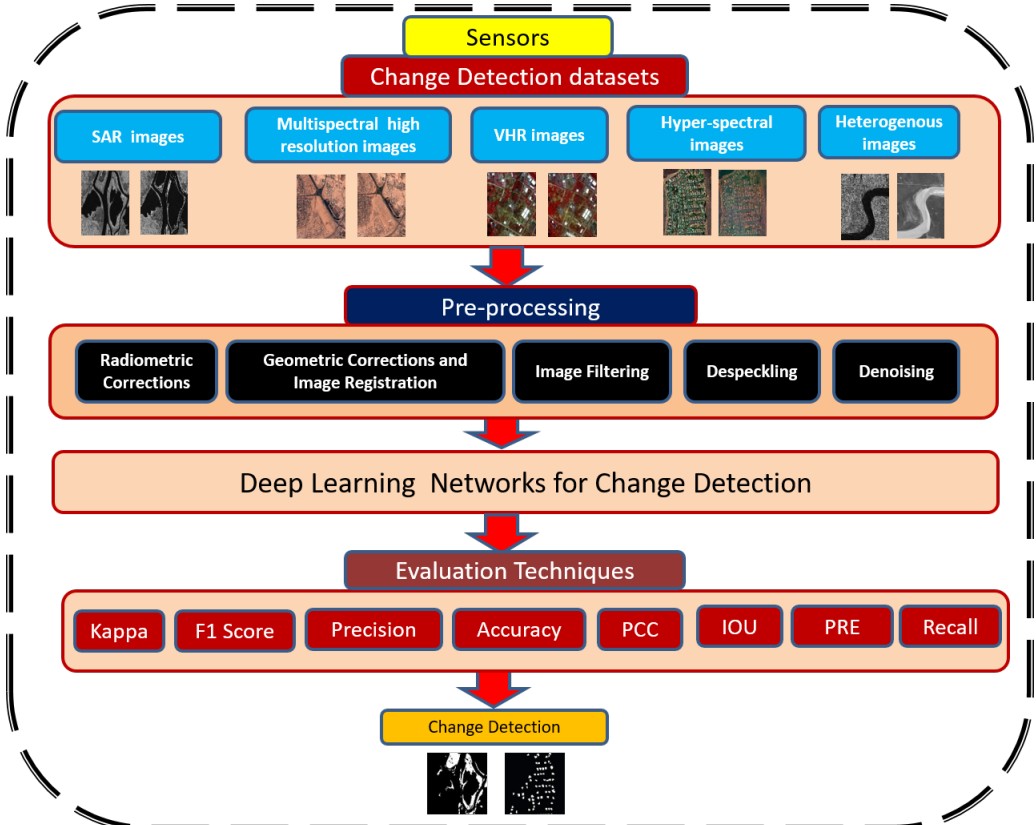

**Figure 3.** The workflow of the change detection architecture.

### 4.1. Pre-Processing

Image pre-processing may have a significant beneficial impact on the quality of feature extraction and image analysis outcomes. The mathematical normalization of a data collection, which is a typical step in many feature descriptor approaches, is akin to image pre-processing. Researchers have created many image processing approaches to solve the problem of atmospheric effects, such as unwanted noise or objects.

### 4.2. Data Collection

Data collection is the first step, and it is critical in identifying changes. The acquisition time of multitude imagery, i.e., season, month, is a critical factor to consider in image collection since it is closely related to phenology, climatic conditions, and solar angle. Therefore, a careful collection of multi-dimensional images is essential to mitigate the consequences of these variables. However, data collection is often constrained by data availability, and the decision is typically a trade-off between the target time, acquisition date, and data availability. The captured data are then grouped into multiple sub-datasets, referred to as multi-temporal images, and sent for pre-processing [100].

### 4.3. Geometric Cegistration

Geometric Registration is a method for detecting unnecessary parts of satellite images to analyze change. It is needed in all change detection techniques because raw aerial imagery involves some degree of geometric distortion. Sensor location variations cause this illusion. Although some distortions are predictable and quickly corrected, others are complex and difficult to exclude from records. Thus, the geometric correction is intended to compensate for distortions and eventually create an image with a high degree of geometric [101,102]. If geometric distortions are not accounted for in images, the spatial coordinates of the pixel would be incorrect. This method is mostly used to identify image changes taken in many dimensions that may cause pixel misclassification. Geometric registration may be performed using RPC, SIFT, DTM, CACO, and RANSAC [103,104]. Aside from these methods, a registration strategy known as Harris-Laplace is used for CD [105,106]. To improve the accuracy of point detection, the identified points are grouped and balanced using SIFT in this method.

### 4.4. Radiometric Correction

Radiometric correction is more effective for optical images. Relative radiometric correction can be used to normalize multitemporal data gathered over distinct periods, eliminating errors or distortions. Although comparing multiple datasets, it was discovered that image enhancement and correction methods play substantial roles. It is used to calibrate the pixel values and to compensate for value errors. The method significantly enhances the interpretability and accuracy of remote sensing data. Calibration and adjustment of radiometric data are critical when comparing different datasets over time. This step employs two distinct approaches: absolute calibration, in which digital numbers are converted to their corresponding ground reflectance values, and relative calibration, which is necessary to calibrate the deflection sensitivity properly [25,107]. The intensity normalization approach is used in satellite images to regulate brightness and contrast. It can be tested by adjusting the histogram of a satellite image as needed. In the digital elevation model (DEM); radiometric corrections also increase the incident-angle oriented surface area [108]. Radiometric corrections that erase geometric distortions remove sensors that produce alterations in scene illumination and geometric corrections.

### 4.5. Despeckling

Log transformations are the general denoising approaches for multiplicative noise. However, the SAR images include more significant noise, necessitating a complicated denoising technique known as despeckling. Despeckling is possible because of the well-designed filters, such as Lee filter, gamma maximum a posteriori, and Kuan filter. SAR images, in addition to Landsat images, are frequently employed for CD. This method increases the performance of the SAR-based CD technology. Among the filters used for despeckling, the Lee sigma filter produced the best results [109]. In the spatial domain, speckle filtering is achieved using an improved Lee sigma filter, while in the temporal domain, accuracy is improved by utilizing temporal similarity between neighboring pixels [110]. Alternatively, the Gaussian noise model, extensively used to remove Gaussian noise, was developed [111]. Rather than observing nearby pixels in an image, pixel values are obtained by assessing the entire image. Moreover, non-linear diffusion filtering is also useful to remove speckle noise in SAR images [112].

### 4.6. Denoising

Image denoising is a fundamental pre-processing process before processing the other tasks, such as segmentation, texture analysis, feature extraction, etc. It is used for various noise reduction tasks, while also retaining image information. Furthermore, a non-local mean filter is used to denoise the SAR image [113]. In [114], a real-time image dimensionality reduction filter is proposed that locates image edges using a thresholding-based method. Moreover, spatial filters work by smoothing across a fixed window, producing

artifacts surrounding the object and occasionally excessive smoothing, resulting in visual blurring. As a result of its qualities, such as sparsity, multiresolution, and multiscale nature, the wavelet transform is ideally suited for performance [115].

## 5. Change Detection in Remote Sensing Datasets by Using DL-Based Networks

In this section, we review various representative CDs in RS images by using DL-based algorithms. We present the literature review of nearly 68 CD papers published from 2015 to 2021 in deep learning. They are classified into different categories (e.g., supervised, unsupervised, and semi-supervised) in each section. To demonstrate the effectiveness of several methods, we present the actual findings for some of them; their advantages and disadvantages are also discussed at the end of each section and in the tables.

### 5.1. SAR Image Change Detection by Using Deep Learning

To demonstrate the performance of several deep learning algorithms for change detection in SAR images, we present the review literature into different DL-based categories, such as supervised, unsupervised, and semi-supervised, in tabular form, in Table 9.

#### 5.1.1. Deep Learning-Based Supervised Methods for SAR Image

Gong et al. [116] presented an innovative CD method using deep learning for multi-temporal SAR images. They trained a deep neural network to generate a CD map directly from the two source images without creating a difference image (DI); the CD problem was simplified as a classification problem. To complete classification, the learning procedure for deep architectures comprises unsupervised feature learning and supervised fine-tuning. Their proposed model consists of three parts as follows: (1) data pre-classification for obtaining high-accuracy data with labels; (2) neural network construction for learning image features and fine-tuning the parameters of the neural network; and (3) for classifying the changed and unchanged pixels using a trained deep neural network. Their suggested method performs well compared to existing methodologies, such as the clustering and thresholding techniques.

Ma et al. [117] provided a unique technique for SAR image CD based on gcForest and multiscale image fusion. gcForest was used since it enhances accuracy and reduces training difficulty. Consequently, the suggested technique uses various sizes of image blocks as gcForest input, allowing it to learn more image attributes while reducing the effect of the image's local information on the classification result. Furthermore, to increase the detection accuracy of pixels with abrupt gray value changes, the suggested technique integrates gradient information from the difference image with the probability map acquired from the well-trained gcForest. As a result, extracting image gradient information may increase the image edge information, and edge detection accuracy can be improved. To reduce the drawback of computational volume and the time-consuming nature of simulation, Samadi et al. [118] used a supervised DL methods for CD in SAR images. They proposed an approach that combines morphological images with two original images to provide a suitable data source for DBN training. The experimental findings showed that the proposed method has sufficient implementation time, desirable performance, and excellent accuracy.

#### 5.1.2. Deep Learning-Based Unsupervised Methods for SAR Image

Gao et al. [119] developed a unique SAR image CD approach using deep semi-NMF and SVD networks. In their suggested method, pre-classification is done using deep semi-NMF and easily obtains high accuracy labeled samples. They presented a CD classification model based on SVD networks. From multi-temporal SAR images, SVD networks can learn nonlinear relations and suppress the noisy unchanging areas. In addition, to improved the classification performance—two SVD convolutional layers were used to get a reliable feature. The suggested approach is unsupervised and does not make any rigorous assumptions.

Planinsic et al. [120] proposed a CD algorithm that extracts features within the tunable Q discrete wavelet transform (TQWT) employing higher-order log cumulates of the fractional Fourier transform (FrFT), which were input into a stacked autoencoder (SAE) to distinguish between changed and unchanged areas. A comparison of SVMs and SAEs was performed. Experiment findings revealed that extracting features inside TQDWT and FrFT produced the most remarkable results, and the SAE outperformed the SVM in terms of accuracy.

The idea proposed in [121] is a deep learning and superpixel feature extraction for CD in SAR image with a contractive autoencoder. The proposed research focused on reducing the performance degradation due to speckle noise. The proposed strategy extracts the features with a stacked contractive autoencoder. The idea presented by Xiao et al. [122] provided a novel image change detection approach. A self-organizing map (SOMDNCD) was presented to produce a successful change map to strike a reasonable balance between noise reduction and the preservation of area edges. First, the approach employs a median filter to enhance the difference image produced by the mean ratio operator, reducing the impact of image point noise on the generation of difference maps. The logarithmic ratio operator produces a more diverse difference map than the logarithmic ratio operator. When used for images, the edge information is substantially preserved, and the rate of missing change is reduced. Second, the network computes a preliminary change map from the difference map, classifies the pixels of the change map based on whether they have changed, and divides the pixels of the change map into three categories: change, noise, and no change. Finally, a DNN is trained with a noise-like training set to remove residual noise in the change class and generate the final change graph. For change detection tasks in different SAR datasets, Dong et al. [123] used multiple convolutional neural networks in which two patches were used as input. They developed a "Siamese samples" network to take patch pairs as input, taking into account the various trade-offs between them. The Siamese sample network uses single branch double samples as discriminative input to construct a binary classifier (i.e., identify changed class and unchanged class). Pseudo Siamese, Siamese, and two-channel networks offer a worse balance between accuracy and run-time than the suggested design and are weak against speckle noise. Moreover, the proposed method uses a reduced algorithm framework compared to the state-of-the-art technique, resulting in less demanding requirements for pre-classification label accuracy.

Bergamasco et al. [124] proposed a convolutional auto-encoders to detect CD in SAR images. They trained the CAE in a unsupervised way. By using a variance-based feature selection approach, this strategy only evaluates the most informative information received by the CAE.

Geng et al. [125] used saliency-guided deep neural networks (SGDNNs), an unsupervised method for five SAR datasets. Their suggested approach can extract potentially changed locations and eliminate background pixels from DI, reducing the impact of speckle-noise on SAR. Hierarchical fuzzy C-means (HFCM) clustering is created to select samples with a greater probability of being changed and unchanged to get pseudo-training samples automatically. Furthermore, to improve sample feature discrimination, DNNs based on non-negative and Fisher-constrained autoencoders are utilized for final detection.

Farahani et al. [126] used a technique based on auto-encoder, which was a deep analysis method used to achieve fused features of SAR, "optical" to benefit from complementary information, to align multi-temporal images by a reduction in spectral and radiometric differences, and made multi-temporal features more similar, for better accuracy in CD.

Saha et al. [127] proposed an unsupervised method, LSTM, and provide a time-series analysis framework that solved the CD issue in a time-series without knowing the event date or any pixel-wise labeled training data.

Shu et al. [128] used a patch-based approach for CD. A mask function converts change labels with irregular shapes into a regular map, allowing the network to learn patches end-to-end. First, SAR images are used to generate training samples and the designed mask. With the presence of a mask, the U-Net-based network can learn end-to-end while

ignoring irregular forms of labels. The newly created change map is processed iteratively to get a new label and mask used in the next learning cycle. Change features are learned iteratively. Through iterative learning, a two-stage update technique improves data variety while suppressing noise.

Qu et al. [129] introduced a unique DDNet. The spatial and frequency domain elements of the DDNet are merged to improve classification performance. They create a multi-region convolution module in the spatial domain to improve the input image patches and increase the central region features. To extract frequency information in the frequency domain, they used DCT and a gating mechanism. Experiments on different datasets revealed that the proposed DDNet outperforms several other CD methods.

### 5.1.3. Deep Learning-Based Semi-Supervised Methods for SAR Image

Gao et al. [130] developed a model for sea ice CD using convolutional-wavelet neural networks (CWNNs). In sea ice change detection, the wavelet transform is employed to minimize speckle noise in SAR images. In their proposed method, dual-tree complex wavelet transform is added into the convolutional neural networks by using CWNN to classify changed and unchanged pixels. Furthermore, a virtual sample production approach is used to generate samples for CWNN training, alleviating the problem of limited samples. The recommended method's efficacy is demonstrated by experimental findings on two SAR datasets.

Wang et al. [131] proposed a semi-supervised DNN for a SAR Image CD with dual-feature representation pixel-wise and context-wise change feature extraction. An LCS-EnsemNet with the label-consistent self-ensemble technique was created specifically for the SAR image CD to address the issues caused by a lack of labeled samples.

**Table 9.** Summary of the literature work of change detection techniques based on a SAR image dataset.

| Author | Year | Techniques | Mode | Advantage | Disadvantage |
|---|---|---|---|---|---|
| Planinsic et al. [120] | 2018 | Stacked autoencoder | Unsupervised | High accuracy | Model complexity |
| Gong et al. [116] | 2015 | DNN | Supervised | High accuracy | High computational complexity |
| Ma et al. [117] | 2019 | gcForest | Supervised | Suppress noise | Rely heavily on the quality of a DI |
| Samadi et al. [118] | 2019 | DBN | Supervised | Time reduction | Limited Label data |
| Gao et al. [119] | 2017 | NMF SVD | Unsupervised | High performance | Non-efficient samples |
| Lv et al. [121] | 2018 | Contractive autoencoder | Unsupervised | High performance | Loss of spatial information |
| Xiao et al. [122] | 2018 | DNN | Unsupervised | Lower missed detection rate | Limited training data |
| Bergamasco et al. [124] | 2019 | CAE | Unsupervised | Doesnot require label data | Not fully suitable |
| Geng et al. [125] | 2019 | DNN | Unsupervised | High performance | Lack of annotation data |
| Farahani, M et al. [126] | 2020 | DA Approach | Unsupervised | Novel | Not used for SAR and optical feature fusion |
| Saha et al. [127] | 2020 | LSTM | Unsupervised | Not require any labeled training sample | Complex |
| Shu et al. [128] | 2021 | U-Net | Unsupervised | High accuracy | Limited training sample |
| Qu et al. [129] | 2021 | DDNet | Unsupervised | effective and Robust | Lack of spatial feature |
| Gao et al. [130] | 2019 | CWNN | Semi-Supervised | Novel method for training data | Potential of the network learning is not fully released |
| Wang et al. [131] | 2021 | LCS-EnsemNet | Semi-Supervised | High efficiency | Computational burden. |
| Dong et al. [123] | 2018 | Siamese samples | Preclassification | Strong speckle noise canceler | High dependency on labeled data |

SAR dataset computational complexities and processing times are the two most significant obstacles to overcome throughout the change detection procedure. An additional issue is speckle noise in SAR images, which has a multiplicative property and always causes an unfavorable impact in all SAR applications. In [116], the authors proposed a change detection method, [119–121], without generating the DI, and was widely used in previous research. Their proposed method exhibits excellent performance on PCC and kappa (0.98, 0.86). The drawback is that the model is complex, with a high computational complexity because of various features and non-efficient samples. Similarly, it achieved better performance of PCC (0.98, 0.99, 0.99) and kappa (0.91, 0.90). However, drawbacks include fewer training samples and classification errors. In [121], the model performance is

high, but it has a loss of spatial information. In [123], by deriving from an incorrect starting change map, the authors obtained accessible labels for training sets. These approaches do not fully use the promise of network learning and prediction. In [125], the novel idea of unsupervised method saliency-guided deep neural networks achieved good performance (PCC 0.99 and kappa 0.92) but lacked annotation data. In reference [118], the authors achieved the highest PCC and kappa results: 0.99, 0.96. The main advantage of their research is a significant reduction in the amount of time required to simulate an algorithm without creating any negative impact on the accuracy, but challenging to produce label data. In [130], a novel virtual sample generation method was used to generate superior and robust samples to the traditional method, and the problem of limited training samples was alleviated. Their proposed approach outperforms PCC and kappa: 0.98, 0.95. In [126], the authors proposed a unique method, domain adaptation (DA) techniques were not utilized in any of the research for SAR or optical feature fusion. However, the model complexity was high, and their result evaluation performance was lower than in previous research. In [128], PCC and kappa was 0.99, 0.94. The main advantage is that model accuracy and computational time is increased. However, training samples are less, and the model may lead to overfitting. In [129], dual-domain CNN was used, including spatial and frequency domains, to detect SAR images more effectively by using this learning-based algorithm. However, they lack the mathematical description and regularization of potential spatial distribution features, which may weaken the feature representation ability of networks. In [131], their proposed model shows high accuracy, but it has a computation burden. In [123], the authors proposed a model that reduces the speckle noise, but the networks will not perform like most existing deep learning change detection approaches.

### 5.2. Multispectral Images Change Detection Using Deep Learning

#### 5.2.1. Deep Learning-Based Supervised Methods for Multispectral Images

Daudt et al. [132] presented two Siamese extensions of fully convolutional networks trained end-to-end from scratch and out-performed the state-of-the-art for CD, both in accuracy and inference speed, without the need for post-processing. Notable among these modifications is the transformation of the fully convolutional encoder–decoder paradigm into a Siamese architecture, which uses skip connections to increase the spatial accuracy of the outputs. Their architectures are more than 500 times faster than earlier approaches.

Mou et al. [133] introduced the recurrent convolutional neural network, a unique network design combining CNN and RNN. It can extract joint spectral–spatial–temporal information and identify change kinds from bitemporal multispectral images. In both visual and quantitative evaluations of experimental results, the suggested model works well.

Zhang et al. [134] used an end–to–end SSJLN for MS images that jointly learned spectral–spatial representations for the CD challenge. SSJLN is made up of three components: spectral–spatial joint representation, feature fusion, and discrimination learning. First, similar to the Siamese CNN, the spectral–spatial joint representation is obtained from the network (S-CNN). Second, the extracted features are fused to represent the different information, adequate for the change detection task. Third, discrimination learning is represented to better investigate the underlying information of generated fused features to depict discrimination.

Lin, Y. et al. [135] proposed a bilateral convolution network to detect changes in bitemporal multispectral images. They trained the model with two symmetric CNNs, which were capable of learning the feature representations. They applied the outer matrix product to the output feature maps to obtain the combined bilinear features. Softmax classifier was applied to produce the change detected results.

#### 5.2.2. Deep Learning-Based Unsupervised Methods for Multispectral Images

Cao et al. [136] developed an approach for generating unique difference images (DIs) for unsupervised CD in multispectral RS datasets. First, they used a DBN to learn local and high-level features from the local neighbour of a given pixel in an unsupervised way.

Second, a BP method was developed to create a DI based on the selected training samples, emphasizing the difference between changed and unchanged regions, while suppressing false changes in unchanged regions. Finally, a simple clustering analysis was used to create the change detection maps. Their suggested method was tested on the "three Rs" datasets and achieved superior performance than traditional standard pixel-level approaches.

Atluri et al. [137] proposed MAU-Net architecture for CD in multispectral HR satellite images. Each block in the framework contains filters that extract features at several resolutions and an attention layer to assist the network in achieving more refined features. The main advantage of MAU-Net is feature propagation and is achieved through residual connections; important feature maps are identified using the attention layer. The Onera dataset was used for comparison purposes. The suggested MAU-Net method achieved state-of-the-art results in change detection.

Gong et al. [138] proposed a GDCN consisting of generative adversarial and discriminating classified networks. The DCN separates the input data into three classes—change, unchanged, and fake class. The DCN may categorize raw image data into two classes when the network is properly trained to produce the final CM. The suggested approach effectively distinguishes change pixels from the unchanged ones.

Saha et al. [139] used a novel method for processing multitemporal images by feeding them separately to a deep network composed of trainable convolutional layers. The training process does not use any external labels, and segmentation labels are derived from the final layer's argmax classification. To detect object segments from individual images and establish a correspondence between distinct multitemporal segments, a novel loss function is used.

Wiratama et al. [140] used a feature-level U-Net to create a robust land cover change segmentation approach in HR multispectral images. The suggested pan-sharpening was introduced by applying a low-pass filter to remove spectral distortion in an IHS high-frequency image. Their proposed network consists of a feature-level subtraction block layer and a U-Net segmentation layer. The feature-level subtraction block layer extracts the dynamic difference image (DI) for all feature levels. CD with HRMS images outperform existing CD algorithms, even under noise, such as geometric distortions and different angles.

Seydi et al. [141], using an end-to-end model, built a CNN-based network with three parallel channels: the first and second extract deep features from the original first- and second-time images, respectively. The third channel is concerned with the extraction of "change" deep features from differencing and "staking" deep features. Each channel also has three types of convolution kernels: 1D, 2D, and 3D-dilated-convolution kernels. The CD map outputs are also analyzed visually and quantitatively by computing nine different accuracy indices. The proposed model has excellent accuracy compared to other traditional methods.

Luo et al. [142] improved the change detection results using deep convolutional generative adversarial and the DeepLabv3+ network. To address the issue that DL networks require many samples and that CD samples are difficult to get, they used both nongenerative and DCGAN generative approaches for data augmentation. The DCGAN network data successfully supplement the sample dataset. Then, to achieve an RS image CD, an SP-DeepLabv3+ was used. The network was upgraded by replacing the deconvolution layer with subpixel convolutions, which increased the network's total accuracy. With sub-pixel convolution, their network predicted RS image change detection. Finally, the generalization performance of their network was tested on different datasets, such as Google Earth, Landsat 8, and OSCD, and showed that the suggested network performs well in terms of generalization.

### 5.2.3. Deep Learning-Based Semi-Supervised Method for Multispectral Images

Alvarez, J. et al. [143] proposed a self-supervised conditional generative adversarial network for multispectral images, trained to generate only the distribution of unchanged samples. Their main idea was to learn the distributions of unchanged samples through

adversarial training to supervise the generator. Their experimental results showed the effectiveness of the proposed model over several CD methods.

Zhang et al. [144] proposed a new FDCNN-based CD approach in which the sub-VGG16 is used to learn deep features from RS images, the FD-Net is used to generate feature difference maps, and the FF-Net is used to fuse these maps by training with a small number of pixel-level samples. Furthermore, to reduce the training time and improve network performance, they created a change magnitude-guided loss function for training based on a cross-entropy loss function, which allows using prior knowledge to reduce pseudo-changes and makes obtaining the final binary change map easier.

Here, we presented some advantages and disadvantages, see Table 10. In [134], the authors achieved the highest accuracy, 0.99, and precision, 0.97, by using the end-to-end SSJLN model. These models are used to circumvent the effect of DI, and several attempts have been made to solve CD tasks in an end-to-end way [132–134,137,138]. However, the drawback is that these end-to-end CNN models require a massive amount of training data. To address these problems, GDCN is used. The generator recovers actual data from the input noise to give more training examples, improve the discriminating classification network's performance, and achieve an accuracy of 0.95. Moreover, the performance is influenced by the clustering algorithm. In [140], the authors handle complicated changes, distortions, and various angle viewing difficulties to offer feature-level U-Net, including a feature-level subtraction block and a U-Net segmentation layer. A pan-sharpening approach was used to improve geometrical and spatial resolutions in addition to controlling minor changes. In [145], a semi-supervised approach was offered to address the lack of correctly-labeled data and obtain an accuracy of 0.84. In [142], the authors utilize both non-generative and DCGAN generative techniques for sample augmentation to address the issue of deep learning networks requiring a large number of samples and change detection samples, being challenging to get. Their model had 0.95 accuracy by using different datasets. In [141], an end-to-end CNN model was used; it did not require any preprocessing and it achieved high accuracy (OA = 0.99 and KC = 0.80), but this model is time-consuming. It was similar for [135], who used a supervised technique. It is challenging to generate labeled data; consequently, unsupervised methods generally outperform supervised ones.

**Table 10.** Summary of the literature work of change detection techniques based on a multispectral image dataset.

| Author | Year | Techniques | Mode | Advantage | Disadvantage |
|---|---|---|---|---|---|
| Daudt et al. [132] | 2018 | FCNN | Supervised | Trained end-to-end | Massive amount of training data |
| Mou et al. [133] | 2018 | RCNN | Supervised | End-to-End | Model could not extract all the deep features. |
| Zhang et al. [134] | 2019 | SSJLN | Supervised | High performance | Large amount of training data |
| Lin, Y et al. [135] | 2020 | BCNNs | Supervised | Trained end to end | Challenging to generate labeled data |
| Cao et al. [136] | 2017 | DBN | Supervised | High accuracy | Processing time |
| Atluri et al. [137] | 2018 | MAU-Net | Supervised | End-to-end | Low performance |
| Gong et al. [138] | 2019 | DCN, GDCN | Supervised | Reduce training sample issue | Model complexity |
| Saha et al. [139] | 2020 | Deep joint segmentation | Unsupervised | Not require any labeled training pixel | Time consuming |
| Wiratama et al. [140] | 2020 | U-Net | Unsupervised | Solve spectral distortion issue | Computational complexity |
| Syedi et al. [141] | 2020 | CNN | Unsupervised | End-to-end | Time consuming |
| Luo et al. [142] | 2020 | DCGAN, DeepLabv3+ | Unsupervised | High performance | Massive amount of training data |
| Zhang et al. [144] | 2020 | FDCNN | Pretrained | strong robustness and generalization ability | Require large pixel-level training samples |
| Alvarez, J et al. [145] | 2020 | S2-cGAN | Semi Supervised | Extract features at multiple resolutions | Model complexity |

## 5.3. Hyperspectral Images Change Detection by Using Deep Learning

### 5.3.1. Deep Learning-Based Supervised Methods in Hyperspectral Images

Syedi, S. et al. [146] proposed a supervised CD method to improve the efficiency of existing CD approaches by combining similarity and distance-based methods. Their

approach demonstrated greater than 96 per cent accuracy, a false alarm rate of less than 0.03, and an area under the curve of around 0.986.

Fandino, J. et al. [147] introduced stacked auto-encoders to extract features from remote sensing HS datasets for multiclass CD. Their experimental results showed the effectiveness of their method while extracting the relevant features of the fused information. The mixed-affinity matrix can efficiently manage with multi-source data at the same time, allowing to learn representative characteristics between the spectra in the GETNET. It not only delivers more plentiful cross-channel gradient information, but it is also an effective method for simultaneous processing of multi-source information fusion.

Hou et al. [148] proposed deep learning algorithms to identify CDs on HRRS images and created a W-Net architecture that recognizes change feature extractions and classifications using stridden convolution, concatenation, and fast connection techniques. Then, using W-Net as a generator, a GAN was developed to train a mapping function that showed the distributions. W-Net and CDGAN both achieved successful outcomes, and CDGAN can exceed W-Net in terms of performance. In contrast to previous approaches, their proposed techniques can get a final CM from the two-source imagery.

Using different hyperspectral image datasets, Moustafa et al. [149] proposed CD architecture known as attention residual recurrent U-Net (Att R2U-Net). This model used four different variants of U-Net, recurrent U-Net, attention U-Net, and attention residual recurrent U-Net. The attention recurrent, the residual model, has the most parameters allocated and performs best for binary and multi-changes. The recurrent U-Net, the residual U-Net model, was suitable for binary and multiclass CD for HSI with its excellent performance. This study supports the notion that DNN can learn complex features and improve HSI CD performance when combined with HSI data.

### 5.3.2. Deep Learning-Based Unsupervised Methods in Hyperspectral Images

Tong et al. [150] developed a unique strategy to resolve the multiple CD problem when few training samples were present in the source image. An unsupervised binary CD was used to create the binary CM. After that, the source image was classified using active learning, and then a classification map of the target image was obtained using the transfer learning method. Finally, post-classification comparison created the multiple CM.

Song et al. [151] proposed Re3FCN, which used LSTM and a 3D convolutional neural network; the approach integrated the advantages of both deep learning-based FCN and convolutional LSTM. The training samples were calculated using the SCA value for each end-way, and the PCA and SCA approaches were coupled to provide highly accurate and reliable samples. This strategy is advantageous for executing CD in conditions where there are no training data. Additionally, Re3FCN (1) extracts spectral—spatial–temporal information from multi-temporal images; (2) effectively identifies binary and multiclass changes while preserving the spatial structural inputs by substituting convolutional layers for fully connected layers; and (3) trained end-to-end.

Saha et al. [152] suggested an unsupervised CD approach for hyperdimensional images that makes use of an untrained deep model as the deep feature extractor. In hyperdimensional images, their proposed method effectively distinguishes changing pixels from unchanged pixels. This article enhances the method for multiple CD by clustering the changed pixels into various groups using a DCV derived from an untrained model.

Syedi et al. [153] introduced a network that consists of the DTW algorithm and CNN. They found a binary CD by using a DTW and Otsu's thresholding technique and multiple CDs obtained by using CNN. Their proposed method had several advantages over the previous method, including (1) high accuracy for both binary and multiple CDs; (2) generating multiple CM using spatial and spectral features; (3) low false alarm rate; and (4) unsupervised framework without training each dataset.

### 5.3.3. Deep Learning Based Semi-Supervised Methods in Hyperspectral Images

Yuan, Y. et al. [154] focused on a semi-supervised CD method to identify the change in HS images under noisy conditions and, thus, proposed a new distance metric learning framework. A regular evolution framework was used to identify changes in a "noisy" environment without eliminating any noise bands, which are influenced by the atmosphere (or water) and are always eliminated manually in other literature. They used a semi-supervised Laplacian regularized metric learning approach, using massive unlabeled data to address the ill-posed sample problems. They performed their proposed methodology on two multi-temporal HS datasets, where they proved to be best under ideal and noisy conditions.

Wang et al. [155] provided a GETNET network for an HS image CD. Their research provides three significant contributions: (1) a mixed-affinity matrix was introduced, which efficiently manages with multi-source data at the same time, allowing to learn representative characteristics between the spectra in the GETNET; (2) a two-dimensional convolutional neural network was designed to learn discriminative features and increase the generalization; and (3) a new hyperspectral image CD dataset was developed for objective comparison.

Huang, F. et al. [156] presented an HS image CD solution based on tensor and DL. They used a tensor-based information model to establish features changed in hyperspectral remote sensing images. They also designed a Boltzmann artificial neural network machine based on the third-order tensor. Using multilayer Tensor3-RBMs, unlabeled data were trained, the BP neural network was replaced with an STM, and a deep belief network with multi-layers was used to improve the accuracy. Their experimental results showed that tensor remote sensing deep belief networks had higher change detection.

Deep learning technology could potentially be used to resolve HSI-based challenges, but there are still some limitations presented in different research studies. We briefly present the research advantages and disadvantage in Table 11. The recurrent, residual U-Net demonstrating the most outstanding performance, in terms of accuracy, is 0.99 [149]. Some of the previous methods also had good performances, as can be seen in references [146–150,155,156]. However, ref. [148] showed the lowest performance—0.64 kappa and a missed alarm rate (MAR) of 0.30. The lower performance involved limited training data, no use of data augmentation, and no pre-trained model used as a backbone. In addition, their proposed method directly obtained the final change from the two images. In [156], TFS-Cube consumed much more time than other approaches due to the extensive usage of DBN deep networks and the TFS-Cube tensor model. TRS-time DBN consumption can be reduced by enhancing the performance of the computer hardware (for example, by adding a GPU module) or by simplifying the algorithm procedure. In [155], pseudo-training sets generated by various CD techniques are necessary to train the network. The algorithm's performance will be affected by the inherent noise in the pseudo-training sets. CNN training has several advantages, one of which is that it simply requires the input image. In [148], the authors' proposed approach was entirely supervised and, hence, less attractive due to the difficulties of human annotation. In [150], the authors used novel methods, improved the multiple change detection problem, and increased accuracy. However, the suggested method's drawback is that the multitemporal image is expected to have the same land-cover types. Thus new land-cover types in the target image cannot be identified. In [151], there is insufficient training data to validate the usage of these networks for low-resolution HSI with a spatial resolution of 30 m.

**Table 11.** Summary of the literature work of change detection techniques based on a hyperspectral image dataset.

| Author | Year | Techniques | Mode | Advantage | Disadvantage |
|---|---|---|---|---|---|
| Syedi, S et al. [146] | 2017 | Similarity based methods | Supervised | High accuracy | Demand for sample data for thresholding |
| Fandino, J et al. [147] | 2018 | SAE ELM or SVM | Supervised | binary and multiclass CD | Training ample issue |
| Hou et al. [148] | 2019 | W-Net | Supervised | Better performance | Training relies on lots of manually annotated data |
| Moustafa et al. [149] | 2021 | ARR-U-Net | Supervised | Both binary and multiclass CD | Computational complexity |
| Tong et al. [150] | 2020 | AL TL | Unsupervised | Multiple CD | New land-cover types in the target image cannot be detected. |
| Saha et al. [152] | 2021 | deep CVA | Unsupervised | Better performance | Time consuming and model complexity |
| Seydi et al. [153] | 2020 | 3D CNN | Unsupervised | Multiclass CD | A lot of training data |
| Yuan, Y et al. [154] | 2015 | SSDM-CD | Semi Supervised | High performance | Not applicable for Spatial information |
| Huang, F et al. [156] | 2019 | TDL | Semi Supervised | Better performance | Only uses spectral feature |
| Wang et al. [155] | 2019 | GETNET | Semi-Supervised | End-to-end 2-D | Training difficulty |
| Song, A et al. [151] | 2020 | Re3FCN, CD | Pretrained | High semantic segmentation result | Insufficient training data |

### 5.4. VHR Images Change Detection Using Deep Learning

5.4.1. Deep Learning-Based Supervised Methods for VHR Images

Peng et al. [157] presented a more advanced U-Net++ design. The dense skip connections of the U-Net++ model were used to learn multiscale feature maps from several semantic layers. They employed a residual block strategy to facilitate the deep FCN network gradient convergence, which was also useful in acquiring more comprehensive information. Furthermore, the MSOF approach was utilized to integrate multiscale side-output feature maps before creating the final change map. They used the weighted binary cross-entropy loss and the dice coefficient loss to successfully decrease the class imbalance impact. Fang et al. [158] used bitemporal VHR images and constructed a hybrid DLSF. The network provides two concurrent streams: DLSFF and Siamese-based CD. The proposed system learns a cross-domain translation using unique change map references to hide the differences of unchanged areas and emphasizes the differences of changed regions in two domains, respectively. It then concentrates on detecting change regions. Chen et al. [159] proposed two unique deep siamese convolutional neural networks based on MFCU for unsupervised and supervised change detection. DSMSCN is trained on data generated by automated preclassification in unsupervised change detection. DSMS-FCN is capable of processing imagery of any size and does not require a sliding patch-window in supervised change detection, therefore accuracy and inference time might be considerably improved. To address the issue of inaccurate localization, the FC-CRF is used to modify the DSMS-FCN findings. The FC-CRF is integrated with DSMS-FCN by using the output of DSMS-FCN as unary potential.

Jing, R et al. [160] developed a unique DL architecture for a CD consisting of a subnetwork and an LSTM sub-network that used spatial, spectral, and multiphase information to increase the CD capability in VHR RS images. The experiments revealed that the multiphase information extracted by the LSTM sub-network was essential for improving the accuracy of CD results.

5.4.2. Deep Learning-Based Unsupervised Methods for VHR Images

Detecting changes in very high-resolution (VHR) is extremely challenging owing to the effects of seasonal variations, imaging conditions, etc. Javed et al. [161] proposed an object-based CD method for VHR images. They generated MBI feature images and used three different methods of PBCD and proposed a D–S theory theory for detecting the building CD. Correa et al. [162] employed the CD method VHR image. The integrated technique, which consists of two principles, deals with multispectral and multitemporal data captured by different sensors. (i) spectral, radiometric, and geometric homogenization of images obtained by different sensors; and (ii) detection of numerous changes through features that ensure homogeneity across time and between sensors. The main idea is to transform images into some common features using transformation. In this paper, for example, the tasselled

caps transform is used for image transformation. Experiments with various multispectral sensor data were used to evaluate the algorithm.

Saha et al. [163] introduced a deep change vector analysis (DCVA) for VHR image CD, developed by combining CVA with a pre-trained deep convolutional neural network. Deep features are extracted from a pre-trained multilayer CNN. A feature hype vector is formed by combining features from different layers of CNN to ensure that the spatial context is captured at multiple levels of abstraction. Pixel comparisons of deep change vectors from pre-change and post-change images yield deep change vectors, which are then analyzed to extract binary and multiple-change information from multitemporal VHR images.

Zhao et al. [164] proposed an attention gates generative adversarial adaptation network (AG-GAAN) .The AG-contributions GAAN's are as follows: (1) This method can detect multiple changes automatically; (2) it includes an attention gates mechanism for spatial constraint and accelerates change area identification with finer contours; and (3) the domain similarity loss is introduced to improve the model's discriminability, allowing the model to more accurately map out real changes.

### 5.4.3. Deep Learning-Based Semi-Supervised Methods for VHR Images

Saha et al. [165] used a graph convolutional network (GCN) that recently showed good performance in semi-supervised single date analysis to improve change detection performance. To process the parcels into a graph representation that can be handled by GCN, a novel graph construction approach is applied. GCN optimizes its loss function solely on labeled parcels. The iterative training method aids in the propagation of label information from labeled nodes to unlabeled nodes, allowing to detect changes in unlabeled data. The suggested method is based solely on the analyzed bitemporal scene and does not require other datasets or pre-trained networks.

Pang et al. [166] suggested a novel Siamese correlation-and-attention-based CD network (SCA-CDNet) for bitemporal VHR images. Data augmentation was used to successfully prevent overfitting and increase the training model's generalization capabilities in the first stage. Second, ResNet was used to extract the image's multiscale features and fully use the network's current pretraining weights to make subsequent model training easier. Third, a novel correlation module is being developed to consistently stack the aforesaid bitemporal characteristics and extract change features with reduced dimensions. Fourth, an attention model is included between the correlation and decoder modules, causing the network to pay more attention to areas or channels that have an enormous impact on change analysis. Fifth, a novel weighted cross-entropy loss function was developed, allowing training to focus on error detection and improving the training model's ultimate accuracy.

Papadomanolaki et al. [167] proposed a unique model using fully convolutional LSTM networks and presented a U-Net-like architecture (LU-Net) that models the temporal relationship of spatial feature representations by layering integrated fully convolutional LSTM blocks on top of each encoding level and with an additional decoding branch that performs semantic segmentation on the available semantic categories presented in the various input dates, resulting in a multitask framework.

In reference [166], the authors proposed method exhibits excellent performance on accuracy OA = 0.990. F1 = 0.91, Pre = 0.92, and IOU = 0.83. Some previous methods have shown good performance [157–159,162], but there exists some challenges. All of the above deep learning algorithms have one thing in common: they all require a lot of training data, as mentioned in Table 12. In deep learning, it is commonly known that generalization to new images suffers greatly if the training data are insufficient. In [162], the authors mentioned a few future ideas for addressing some of the paper's limitations at the end, such as associating clusters with specific sorts of changes and feature selections to further separate different changes. An improved U-Net++ model with novel deep supervision was presented to capture subtle changes in challenging scenes. According to the authors, the model focuses solely on change/no-change information, which is insufficient for some practical applications [157].

**Table 12.** Summary of the literature work of change detection techniques based on the VHR image dataset.

| Author | Year | Techniques | Mode | Advantage | Disadvantage |
|---|---|---|---|---|---|
| Peng et al. [157] | 2019 | U-Net++ | Supervised | End-to-end | Require huge training sample |
| Fang et al. [158] | 2019 | DLSF | Supervised | High detection performance | Not focus on spectral information changes. |
| Chen et al. [159] | 2019 | SiamCRNN | Supervised | High performance | Large number of labeled sample |
| Jing, R et al. [160] | 2020 | TriSiamese LSTM | Supervised | Improved accuracy | Computational complexity |
| Javed et al. [161] | 2020 | D–S theory | Unsupervised | Low false alarm | Miss detection of building changes |
| Correa et al. [162] | 2018 | Tree of radiometric change | Unsupervised | Good performance | Lot of training sample |
| Saha et al. [163] | 2019 | Multi-layered CNN | Unsupervised | Reduce dependence on changing samples | Needs a large number of pixel-level samples. |
| Zhao et al. [164] | 2020 | AG-GAAN | Unsupervised | Improve the detection accuracy | Model is greatly challenged by the hazardous environments |
| Papadomanolakiet al. [167] | 2021 | LU-Net | Unsupervised | Novel method | Low performance |
| Saha et al. [165] | 2020 | GCN | Semi-supervised | Eliminates many redundant features | Time consuming |
| Pang et al. [157] | 2021 | SCA-CDNet | Pretrained | Improve accuracy | Insufficient for some practical applications |
| Ji et al. [168] | 2019 | Mask R-CNN, CNN | Self-trained | Reduce the demand of training samples | Time complexity |

*5.5. Heterogeneous Images Change Detection by Using Deep Learning*

5.5.1. Deep Learning-Based Supervised Methods for Heterogeneous Images

Yang et al. [169] provided a unique cross-sensor CD approach based on deep canonical correlation analysis (DCCA). Following training with samples from the entire area, the DCCA transformation allows aligning the spectrum of two heterogeneous multi-spectral datasets; then, any change detection approach is used. Experiments on commonly used cross-sensor image datasets show that the suggested strategy outperforms previous approaches. Furthermore, the parametric form of DCCA is often faster to train than the non-parametric form of KCCA.

Wang et al. [170] used a supervised CD method based on the deep Siamese convolutional network with a hybrid convolutional feature extraction module (OB-DSCNH) for multisensor datasets. The suggested approach may extract hierarchical features from input image pairings that are more abstract and resilient than comparing methods.

Ebel et al. [171] proposed a novel Siamese network and suggested a new bimodal fusion-based CD model that combines data from both SAR and optical sensors.

5.5.2. Deep Learning-Based Unsupervised Methods for Heterogeneous Images

Liu et al. [172] proposed a SCCN to reduce the limitations of 1D and 2D CNNs for CD. Their suggested SCCN model is entirely unsupervised, with no labeled pixels. SCCN has one convolutional layer and multiple coupling layers on each side that turn the two input images (fed to each side) into a feature space with more consistent feature representations for the two input images. Finally, the difference map is generated directly in this feature space using pixel-wise Euclidean distances. A coupling function is developed to drive network parameter learning. They pre-train the network layer-by-layer using DAE while taking the noise models of the two input images into consideration to give correct initialization for both network parameters and unchanging labels. The idea proposed by Niu et al. [173] used a cGAN to convert the heterogeneous SAR and optical images into a space where their information is more consistently represented, allowing for direct comparison. The proposed framework includes a cGAN-based translation network that attempts to translate the optical with the SAR image as a target and an approximation network that reduces the pixel-wise gap between the SAR image and the translated one. The two networks are updated alternately. When adequately trained, the two translated and estimated images may be deemed homogenous, allowing direct comparison of the final change map.

Zhan et al. [174] proposed a unique approach to a logarithmic transformation feature learning (LTFL) network to convert the SAR image to the optical image. The modified image pair can then be used to learn high-level feature representations using joint feature extraction. The pre-classification result will be raw data, to pick labeled samples, for training a primary neural network classifier. When this classifier is adequately trained, it will

label each position, thereby identifying changes on the ground. To solve the binary classification problem between heterogeneous pairs of RS images, Touati et al. [175] proposed a stacked sparse autoencoder unsupervised method and trained the temporal image features. The constructed anomaly detection model reconstructs the input from its representation in the latent space to identify pixels of new unseen image pairs. First, a stacked hidden representation is used to encode the probing (new) image (i.e., the bitemporal heterogeneous image pair as the input request) in this normal compact space. The reconstruction error is calculated in the residual normal space using the L2 norm, in which modest reconstruction errors distinguish non-change patterns as belonging to the normal class. In contrast, change patterns are distinguished by high reconstruction errors, as belonging to the abnormal class. The changed/unchanged classification map is produced in the residual space by grouping the reconstructed errors using a Gaussian mixture.

Jiang et al. [176] used a new DHFF method for detecting changes in the heterogeneous image. The suggested deep homogeneous feature fusion approach considers the homogeneous transformation, which converts heterogeneous images into the same feature space as an IST issue. In contrast to the standard IST methodology, which transfers image styles, the proposed DHFF method measures and then achieves feature homogeneity in additional new feature subspaces, using the IIST strategy to fulfill the feature homogeneity requirements for CD in homogeneous images.

Prexl et al. [177], used an unsupervised CD approach and extended DCVA where pre-change and post-change imagery were obtained with differing spatial resolutions and spectral bands.

Sun, Y et al. [178] developed a CD approach focused on image similarity measurements in heterogeneous images. The approach generated a graph for each patch based on a nonlocal patch similarity to create a link between heterogeneous data and then measured the change level by assessing how much the graph structure of one image still confronted that of another image. The graph structure was compared in the same domain, the leakage of heterogeneous data were avoided, resulting in more robust change detection findings. Experiments show that the suggested nonlocal patch similarity-based heterogeneous CD approach works well.

For detecting changes in heterogeneous RS images, Li et al. [179] proposed SSPCN analyses, two heterogeneous images in a high-dimensional feature space, completely unsupervised, with no explicitly labeled examples. A classification-based method was used to establish the pseudo-labels in the proposed method, and each sample was provided a weight to reflect the ease of the sample. Then, SPL was used to learn simple samples initially and then gradually incorporate more detailed data. During the training process, the sample weights were dynamically adjusted based on the network parameters. Finally, a trained convolutional neural network was used to build a binary change map.

Yang et al. [180] developed a new selective adversarial adaptation method for SAR images. The major contribution was to transfer knowledge from different source domains to aid in the identification of changes in the target domain. In their proposed method, they first used a discriminator to choose a sample that fit in the target domain and another discriminator was used to minimize the domain discrepancy by adversarial learning.

5.5.3. Deep Learning-Based Semi-Supervised Methods for Heterogeneous Images

Wu et al. [181] proposed a semi-supervised CD strategy based on GCN and a multiscale object-oriented analysis to solve CD problems better (in homogeneous and heterogeneous). Their proposed approach first performs image segmentation, then constructs image blocks into a graph, and uses GCN to detect which blocks are changed.

Jiang et al. [176] used a new DHFF method to detect changes in the heterogeneous image. The suggested deep homogeneous feature fusion approach considers the homogeneous transformation, which converts heterogeneous images into the same feature space as an IST issue. In contrast to the standard IST methodology, which transfers image styles, the proposed DHFF method measures and then achieves feature homogeneity in addi-

tional new feature subspaces using the IIST strategy to fulfill the feature homogeneity requirements for CD in homogeneous images.

Saha et al. [182] presented a unique self-supervised learning method for CD in a bitemporal scene; they used different concepts of self-supervised learning literature, such as deep clustering, augmented view, contrastive learning, and the Siamese network, while only utilizing the available target unlabeled scene. Their proposed method can train a network that can effectively exploit these concepts and modify them appropriately for the target multisensor bitemporal data.

Various methods show high performances in [169,172–176,178,179]. However, there are still some limitations, as presented in Table 13. In [172], the authors show the method's superiority over several existing approaches. This type of method relies on pre-classification and does not need labeled data. However, it is still incapable of utilizing an enormous amount of remote sensing data that are currently available and can be used to improve performance. Furthermore, the method's limitation is that it only considers the unchanged pixels. In [173], fully linked layers that receive pixels as input reduce pixel-wise variances in their networks. These types of model networks do not take the neighborhood information around a pixel, and they have many learnable parameters—connected layers that accept a pixel as input. Furthermore, the selections of training samples are dependent on naive existing approaches. Thus, the subsequent process may be harmed by resulting errors.

**Table 13.** Summary of the literature work, of change detection techniques based on the heterogeneous image dataset.

| Author | Year | Techniques | Mode | Advantage | Disadvantage |
|---|---|---|---|---|---|
| Yang et al. [169] | 2018 | DCCA | Supervised | DCCA typically faster to train than KCCA | High computational cost |
| Wang et al. [170] | 2020 | OB-DSCNH | Supervised | High accuracy | Did not consider if central pixel and its neighborhoods are not in the same category |
| Ebel et al.[171] | 2021 | Siamese network | Supervised | Novel data | Time consuming and low performance |
| Liu et al. [172] | 2016 | SCCN, DAE | Unsupervised | High performance | spatial complexity |
| Niu et al. [173] | 2018 | cGAN | Unsupervised | Higher accuracy | Huge amount of learnable parameters |
| Zhan et al. [174] | 2018 | LTFL | Unsupervised | High detection accuracy | High cost of manual operation |
| Touati et al. [175] | 2020 | DSRM | Unsupervised | better performance | Require huge training sample |
| Saha et al. [182] | 2021 | DC,AV,CL,SN | Self-supervised | Better performance | Time consuming |
| Yang et al. [180] | 2021 | SAA | Unsupervised | Novel and high performance | Training is difficult |
| Li et al. [179] | 2021 | SSPCN | Unsupervised | Better accuracy | Generation of Pseudo labels does not hold in some case , |
| Prexl et al.[177] | 2021 | Extended DCVA | Unsupervised | Better performance | Not Novel |
| Sun et al. [178] | 2021 | Patch similarity | Unsupervised | Better performance | Complex when the ground features covers a very large area |
| Wu et al. [181] | 2021 | GCN | Semi-Supervised | Novel framework | Time consuming |
| Jiang et al. [176] | 2020 | DHFF IST | Pre-trained | High performance | Computational complexity |

In [169], the key limitation is the high computational costs. In [174], despite their high detection accuracies, it requires a high cost of manual operation in practice under supervision mode or a complicated screening process to select training samples under unsupervised mode. In [178], The suggested technique learns a more robust distance-induced probabilistic network adaptively. This local structure consistency uses the fact that heterogeneous images share the same structural information for the same ground object; that is, imaging modality invariant. It mainly focuses on changes in local structures. In [179], the proposed method has several limitations. In the generation of pseudo-labels, it is assumed that multitemporal images have minor changes, and this assumption does not hold in some cases.

One interesting direction that is also new involves planetary CD, in which (according to our knowledge) only a few authors have conducted work. Here, we explain their work:

Kerner et al. [183] developed supervised approaches for planetary CD in depth. For planetary change detection, they used a convolutional autoencoder with various supervised classifiers to detect surface feature changes in a variety of RS datasets, with a small number of labeled training samples. Despite significant differences in image quality, illumination, imaging sensors, surface properties, and co-registration, their proposed method can detect meaningful changes with high accuracy, using relatively small training datasets.

Saha et al. [184] proposed a patch-level unsupervised CD deep transfer-based method for planetary exploration. Their proposed method can determine whether a pair of bitemporal patches are changed and, furthermore, they proposed a technique using pseudo-unchanged pairs to determine the threshold for distinguishing changed and unchanged patches.

## 6. Evaluation Metrics

CD algorithms deal with extremely unbalanced data concerning the ratio of changed regions related to the area that has not been changed. The most commonly used evaluation technique in CD is Accuracy, F1, Precision and Recall, Overall error, per centage of correct classification (PCC), PRE (represents that of expected agreements), and kappa as shown in Equations (1)–(10).

All evaluation metrics are calculated as:

$$Precision = TP/(TP + FP) \tag{1}$$

*TP*: pixels that are identified as change. *TN*: pixel identified as unchanged. *FP*: in the algorithm, unchanged pixels are incorrectly identified as a change. *FN*: change pixels incorrectly classified as unchanged. Precision and recall are two criteria used to assess the efficacy of a retrieval system. Precision is calculated by dividing the number of correct instances obtained by the total number of instances retrieved. Sensitivity is defined as the accuracy calculated as a ratio of reference-changed pixels. It is also known as recall or TPR. Specificity defines the accuracy of unchanged pixels. It is also known as TNR. The ROC curve is plotted using sensitivity and specificity measurements. In addition, the model's performance is determined by the area under the ROC curve (AUC). The best sensitivity and specificity is 1.0, the worst is 0.0. Kappa is commonly used to assess classification performance, with a larger kappa value indicating better performance. F1 is the harmonic mean between precision and recall. It considers both false positives and false negatives and performs well with imbalanced data. Accuracy: it measures how many positive and negative observations are correctly classified [185].

$$Sensitivity/recall = TP/(TP + FN) \tag{2}$$

$$Specificity = TN/(TN + FP). \tag{3}$$

$$F1Score = F1 = 2 * (precision * recall)/(precision + recall) \tag{4}$$

$$Accuracy = Correct\,predictions/Total\,predictions = (TP + TN)/(TP + TN + FP + FN) \tag{5}$$

$$Kappa = (PCC - PRE)/(1 - PRE) \tag{6}$$

$$PCC = (TP + TN)/(TP + TN + FP + FN) \tag{7}$$

$$PRE = ((TN + FP) \cdot (TN + FN))/(TP + TN + FP + FN)^2 + ((TN + FP) \cdot (TN + FN))/(TP + TN + FP + FN)^2 \tag{8}$$

$$Accuracy = (TP + TN)/(TP + TN + FP + FN) \tag{9}$$

$$IOU = TP/(TP + FP + FN) \tag{10}$$

*Quantitative Results*

For assessing the performances of different evaluation matrices, we presented some previous results as an example that are based on change detection by using deep learning methods for different datasets. Table 14 and Figure 4 shows the result by using the SAR dataset. The results were obtained by FCM, NLMFCM, DBN, SCCN, wavelet fusion, gcForest, and the proposed method. Different types of evaluation matrices are used to show the performances, such as FN, FP, OE, PCC, and K. Similarly, Table 15 presents the quantitative results for VHR by using different evaluation matrices, such as precision,

recall, F1 score, and OA; Figure 5 presents the visual comparison of the VHR image. Table 16 presents the quantitative results for hyperspectral images by using different evaluation matrices, such as OA, precision, recall, F1 score, and k; Figure 6 presents the visual comparison of hyperspectral images. Table 17 presents the quantitative results for multispectral images by using different evaluation matrices, such as OA, sensitivity, MD, FA, F1, BA, precision, specificity, and KC. Figure 7 presents the visual comparison of multispectral images. Similarly, Table 18 presents the quantitative results and Figure 8 shows visual comparison for heterogeneous images by using different evaluation matrices, such as FA, MA, OE, OA, and KC.

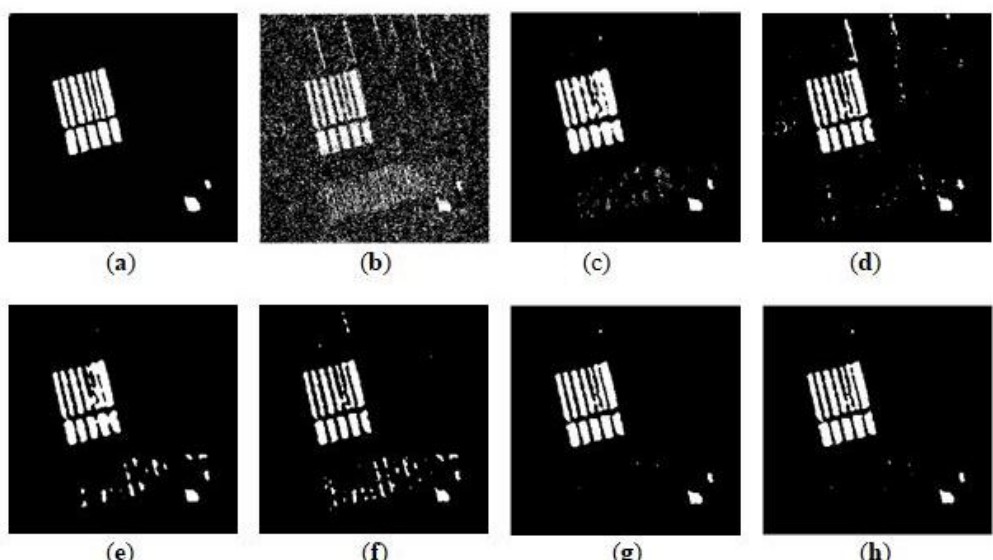

**Figure 4.** Visual comparison of CD results using various methods for Farmland C: (**a**) reference, (**b**) FCM, (**c**) NLMFCM, (**d**) DBN, (**e**) SCCN, (**f**) wavelet fusion, (**g**) gcForest, (**h**) proposed method.

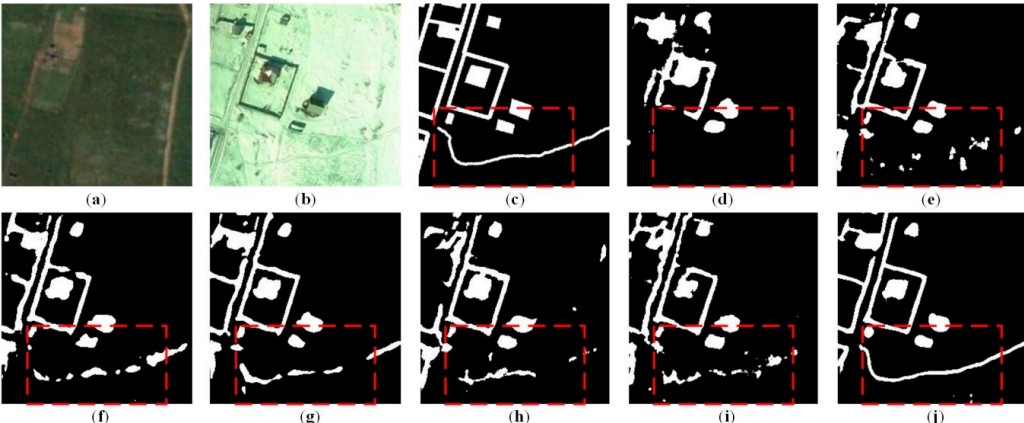

**Figure 5.** Visual comparison of CD results using various DL methods for area 6: (**a**) image T1, (**b**) image T2, (**c**) reference change map, (**d**) CDNet, (**e**) FC-EF, (**f**) FC-Siam-conc, (**g**) FC-Siam-diff, (**h**) FC-EF-Res, (**i**) FCN-PP, and (**j**) U-Net++. The changed parts are marked in white while the unchanged are in black.

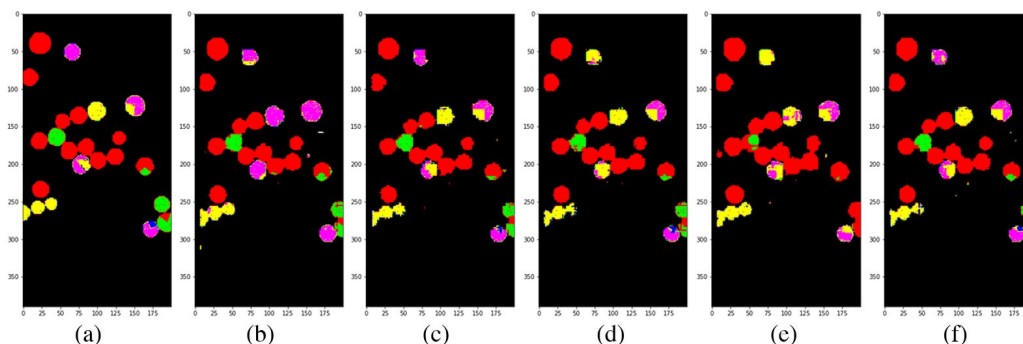

**Figure 6.** CD results for hyperspectral hermiston images (**a**) ground-truth, (**b**) U-Net, (**c**) RU-Net, (**d**) Att U-Net, (**e**) R2U-Net, and (**f**) Att R2U-Net.

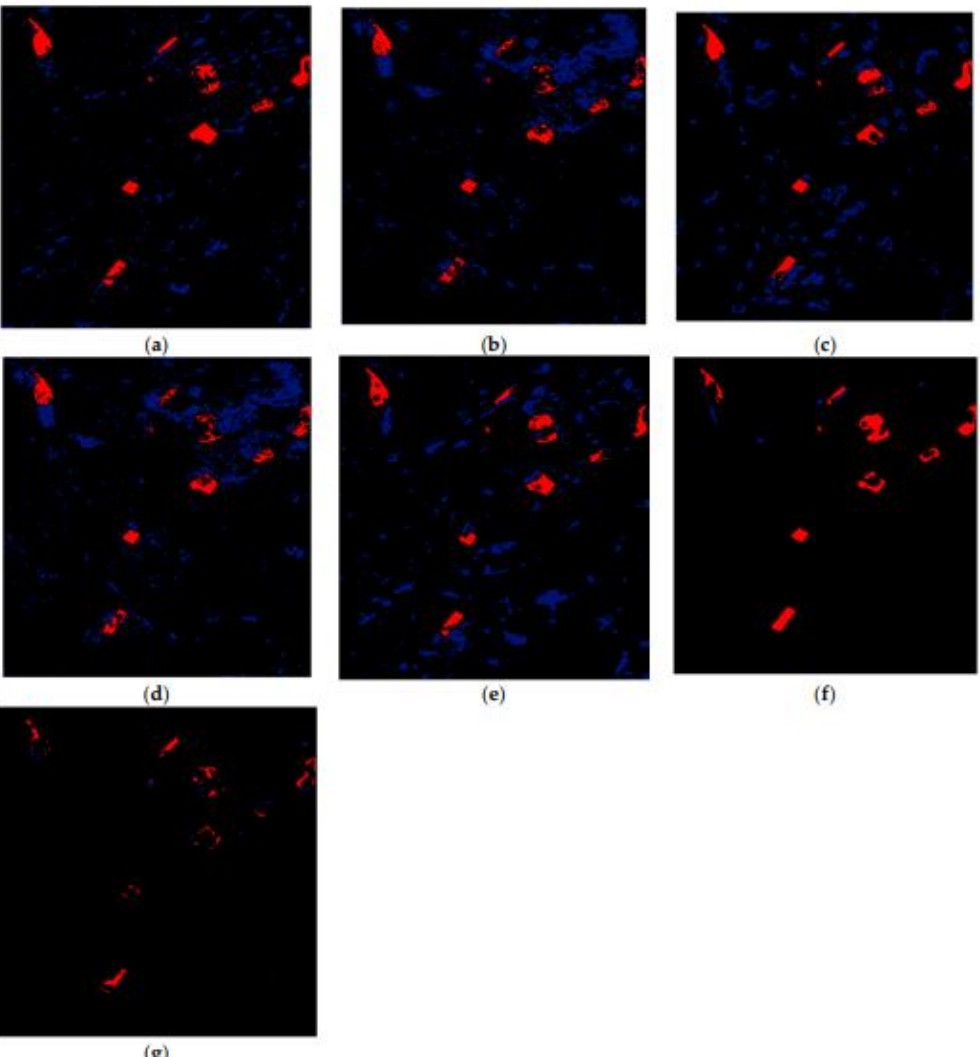

**Figure 7.** CD results for multispectral-Saclay images (**a**) CVA-SVM, (**b**) MAD-SVM, (**c**) PCA-SVM, (**d**) IR-MAD-SVM, (**e**) SFA-SVM, (**f**) 3D-CNN, and (**g**) Proposed Method. Black, red, and blue colors indicate TP and TN, FN, and FP pixels.

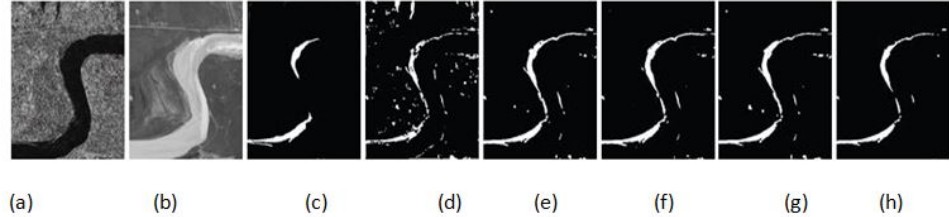

**Figure 8.** CD results for the heterogeneous images. (**a**,**b**) Images acquired by different observation times, (**c**) reference image. (**d**) PCC, (**e**) SCCN, (**f**) ASDNN, (**g**) FL-based, (**h**) LTFL.

**Table 14.** Quantitative results for SAR images [117].

| Methods | FN | FP | OE | PCC | K |
|---|---|---|---|---|---|
| FCM | 12,126 | 813 | 12,939 | 85.47 | 34.95 |
| NLMFCM | 687 | 668 | 1355 | 98.48 | 86.36 |
| DBN | 697 | 841 | 1538 | 98.27 | 84.29 |
| SCCN | 768 | 779 | 1547 | 98.26 | 84.38 |
| wavelet fusion | 931 | 1377 | 2308 | 97.41 | 75.76 |
| gcForest | 124 | 685 | 809 | 99.09 | 91.41 |
| Proposed method | 163 | 630 | 793 | 99.11 | 91.66 |

**Table 15.** Quantitative results for VHR images [157].

| Methods | Precision | Recall | F1-Score | OA |
|---|---|---|---|---|
| CDNet | 0.7395 | 0.6797 | 0.68 82 | 0.9105 |
| FC-EF | 0.8156 | 0.7613 | 0.7711 | 0.9413 |
| FC-Siam-conc | 0.8441 | 0.8250 | 0.8250 | 0.9572 |
| FC-Siam-diff | 0.8578 | 0.8364 | 0.8373 | 0.9575 |
| FC-EF-Res | 0.8093 | 0.7881 | 0.7861 | 0.9436 |
| FCN-PP | 0.8264 | 0.8060 | 0.8047 | 0.9536 |
| U-Net++ | 0.8954 | 0.8711 | 0.8756 | 0.9673 |

**Table 16.** Quantitative results for hyperspectral images [149].

| Methods | OA | Precision | Recall | F1-Score | K |
|---|---|---|---|---|---|
| U-Net | 0.945470 | 0.935675 | 0.951087 | 0.942151 | 0.950427 |
| R U-Net | 0.989402 | 0.948417 | 0.923722 | 0.935821 | 0.945470 |
| Att U-Net | 0.986232 | 0.900143 | 0.908169 | 0.893870 | 0.930937 |
| R2 U-Net | 0. 953387 | 0.978676 | 0.920067 | 0.919009 | 0.900139 |
| Att R2U-Net | 0.991611 | 0.958538 | 0.946333 | 0.952342 | 0.957096 |

**Table 17.** Quantitative results for multispectral images [157].

| Methods | OA | Sensitivity | MD | FA | F1 | BA | Precision | Specificity | KC |
|---|---|---|---|---|---|---|---|---|---|
| CVA | 94.09 | 19.50 | 80.50 | 4.16 | 13.15 | 57.67 | 9.92 | 95.84 | 0.104 |
| MAD | 91.05 | 42.48 | 57.52 | 7.81 | 17.88 | 67.34 | 11.32 | 92.19 | 0.148 |
| PCA | 92.55 | 19.25 | 80.75 | 5.72 | 10.61 | 56.77 | 7.32 | 94.28 | 0.075 |
| IR-MAD | 91.1 | 40.56 | 59.44 | 7.70 | 17.31 | 66.43 | 11.00 | 92.30 | 0.142 |
| SFA | 92.48 | 31.06 | 68.94 | 6.07 | 15.94 | 62.49 | 10.72 | 93.93 | 0.128 |
| 3D-CNN | 98.15 | 29.19 | 70.81 | 0.23 | 42.02 | 64.48 | 74.96 | 99.77 | 0.413 |
| Proposed method | 99.18 | 75.40 | 24.60 | 0.25 | 80.99 | 87.58 87.46 | 99.75 | 0.805 | |

**Table 18.** Quantitative results for heterogeneous images [174].

| Methods | FA | MA | OE | OA | KC |
|---|---|---|---|---|---|
| PCC | 2947 | 1187 | 4134 | 95.86 | 0.4651 |
| SCCN | 2094 | 538 | 2632 | 97.36 | 0.6532 |
| ASDNN | 1939 | 525 | 2464 | 97.53 | 0.6695 |
| FL-based | 2027 | 627 | 2654 | 97.34 | 0.6434 |
| LTFL | 1104 | 841 | 1945 | 98.05 | 0.6950 |

## 7. Discussion

There are numerous applications on deep learning methods for CD in remote sensing, and deep learning-based techniques have proven to be highly successful. However, there are numerous problems in the processes, and they are illustrated as follows:

### 7.1. Training Sample

Although DL algorithms may learn highly abstract feature representations from raw remote sensing images, detection and identification success is dependent on a large number of training samples. However, because collecting labeled high-resolution images is challenging, there is (frequently) a scarcity of high-quality training. Under these conditions, retaining the representation learning performance of DL algorithms with fewer appropriate training samples remains a significant problem. To properly train systems, DL researchers employ a variety of strategies, including transfer learning [186–192], data augmentation, GAN [148,155], AE, and SSAE [193]. While these techniques alleviate some of the problems associated with a lack of samples, further improvement is needed. One DL method [194] involves trusting small training datasets for supervised CD. This approach appears to be quite interesting as it minimizes the requirements of labeled training data.

### 7.2. Prior Knowledge

Due to the change location and direction ambiguity, the total area on the change map is greater than the changing area in the change detection. Due to a lack of prior information, conventional unsupervised approaches are unable to rapidly solve this. Thus, weak and semi-supervised approaches are used, but advanced study is required to increase their performances.

### 7.3. Image Registration

In recent years, deep learning RS image registration has become an active research area. The main limitation of deep learning in image registration is the lack of available public training datasets, which should be a future endeavor of the remote sensing community. There are many challenges to be addressed, especially in remote sensor datasets. Due to

the diversity of remote sensing images acquired at different resolutions and at different times (or by different modalities), it will be an important challenge and laborious task to establish huge public training datasets for image registration [163]. Non-nadir image registration [50], for example, from rovers, satellites, or airborne sensors, remain unresolved challenges. The use of robust prediction algorithms to deal with registration errors is one possible solution to the problem. Some algorithms have been proposed to reduce the registration errors [195]. Moreover, accurate registration of remote sensing imagery, such as multitemporal images, is very challenging. To better understand how to minimize the influence of residual misregistration on the change detection process, ref. [196] investigated the behaviors of registration noise that affect multitemporal VHR datasets. More research on robust and reliable registration methods is required.

*7.4. Rs Image Complexity*

RS data are complex due to the various radiation and scattering characteristics of visible light, microwave, and infrared; image behavior varied greatly with diverse ground features. In contrast to natural scene images, high-resolution remote sensing images contain a variety of objects with various colors, sizes, rotations, and locations in a single scene. In contrast, unique scenes from other categories may resemble each other in many ways. The complexity of RS images significantly contributes to the challenge of learning stable and discriminative representations from scenes and objects using DL. The RS community recognizes that the basic problems of distant observations will never be solved. Separating data from noise to recover a specific set of geophysical characteristics, for example, and precise sensor calibrations are ongoing issues. Technological developments enhance the information content of the observations, but the data are never entirely sufficient to uniquely identify all of the geophysical characteristics of interest; the list of needed "observables" expands inexorably with scientific progress. As a result, RS remains a fundamentally ill-posed issue that must be properly characterized and limited by theoretical models, prior knowledge, and auxiliary observations. These are crucial factors to consider while developing new scientific aims. In [197,198], the authors present some challenges and a few helpful suggestions to overcome RS problems, which are helpful for researchers.

*7.5. Multiple Change Maps*

Most CD algorithms only identify binary changes and neglect multiple change detections in remote sensing images, mostly researches are concentrated on binary CD, and only discriminate between the presence and absence of change. There is limited work on multiple CDs, in which the change class is further subdivided into several types of change. Few studies, such as [149,150,152,153,163], have focused on multiple change maps. Furthermore, hyperdimensional images are used for binary CDs; to the best of our knowledge, only one study has used hyperdimensional images, i.e., [152], for multiclass CD. Thus, there is a need for further research. Scholars should introduce a new method for multiple change maps on different datasets, because in the past, some researchers only used hyperspectral images for multiple CDs, so there is also a need for more experiments on other RS images, such as VHR images and hyperdimensional images.

## 8. Conclusions

We reviewed some of the most well-known RS datasets and the latest DL algorithms for change detection in the literature that achieved outstanding performances. In addition, a deeper review was conducted to describe and discuss the use of DL algorithms in its categories, such as supervised change detection methods, unsupervised change detection methods, and semi-supervised methods; we also presented their advantages and disadvantages, which differentiates our study from previous reviews. The systematic analysis and commonly used networks in DL adopted for change detection show that great progress has been made for change detection, but there are still many challenges in CD due to a lack of training data, prior knowledge, image complexity, etc. Nevertheless, even if these

challenges are overcome, due to evolving demands and diverse data, there are still many core issues in RS datasets that have been not focused on as yet, such as heterogeneous data, multiresolution images, and global information of high-resolution and large-scale images. Therefore, further studies with more focus on these challenges is strongly suggested.

**Author Contributions:** All of the authors made significant contributions to the article. The review was created and written by A.S. under the supervision of G.C. Z.K. and M.A. (Muhammad Asad); M.A. (Muhammad Aslam) helped with the review of the related literature. The general framework of the article was discussed by all contributors. A.S. and G.C. created the overall framework of the review, examined the articles, and oversaw the study throughout all stages. All authors have read and agreed to the published version of the manuscript.

**Funding:** This study was funded in part by the Jiangsu Provincial Natural Science Foundation under grant BK20191284 and the National Natural Science Foundation of China under grant 61801222.

**Acknowledgments:** The authors truly appreciate the helpful remarks and constructive ideas provided by the academic editors and reviewers.

**Conflicts of Interest:** The authors declare no conflict of interest.

## Abbreviations

The following abbreviations are used in this manuscript:

| | |
|---|---|
| RPC | rational polynomial coefficient |
| DTM | digital terrain model |
| SIFT | scale-invariant feature transform |
| PSOSAC | particle swarm optimization sample consensus |
| CACO | continuous ant colony optimization |
| RANSAC | random sample consensus |
| DCGAN | Deep convolutional generative adversarial network |
| SSJLN | spectral–spatial joint learning network |
| HRMS | high-resolution multispectral |
| SCCN | symmetric convolutional coupling network |
| DHFF | deep homogeneous feature fusion |
| MFCU | multiscale feature convolution unit |
| DSMS-CN | deep Siamese multiscale convolutional network |
| FCN | fully convolutional network |
| DLSF | dual learning-based Siamese framework |
| FC–CRF | fully connected conditional random field |
| KCCA | kernel canonical correlation analysis |
| DCV | deep change vector |
| IST | image style transfer |
| DHFF | deep homogeneous feature fusion |
| SSPCN | spatially self-paced convolutional network |
| SPL | self-paced learning |
| TPR | true positive rate |
| TNR | true negative rate |
| ROC | receiver operating characteristic |

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
