# Peer review of "Deep Learning-Based Change Detection in Remote Sensing Images: A Review"

_remotesensing, doi:10.3390/rs14040871_

Round 1

Reviewer 1 Report

Change detection is an important research area in remote sensing. This manuscript provides a survey on this topic, especially focusing on 
deep learning based methods. While the manuscript can be an interesting addition to the change detection literature. However, few Sections are
incomplete. Organization is not proper in few Sections. Furthermore, there are some errors at some places. Thus,  revision/improvement 
is required in several sections. Here are more detailed comments:

1) Introduction: add a paragraph stating how is this survey different from the previous existing surveys.

2) Section 2: should be moved towards the end of the manuscript, i.e., after describing the relevant methods.

3) Furthermore, some contents of Section 2 is irrelevant and should be excluded from this manuscript. E.g., Table 1 for which readers can 
refer to other surveys. Adding to this, there are some errors in this Table. Sentinel 6 is listed, however Sentinel 1 or Sentinel 2 is not listed.
Worlview 4 is listed, however Worldview 2 is not listed.

4) Table 2: illustration of SAR CD datasets - present such datasets while discussing each sensor separately.

5) Table 2: Please pay attention that "European space" is not a satellite name.

6) Table 2: Landsat 7 is not a SAR sensor.

7) Table 2: how is Radarsat SAR different from Radarsat-2 ?

8) Line 212: "occasional targets." meaning not clear.

9) Line 214: "the internal details of the change" - meaning not clear.

10) Line 227: Please define what is unsupervised annotation method?

11) Table 3: Google Earth is not a valid satellite name. You can write it as unknown.

12) Table 3: Onera dateset needs to be added here.

13)  Table 4: what is HIS sensor?

14) Table 5: Google Earth is not a valid sensor.

15) Please remove Figure 3 - it is not adding any information.

16) I will suggest to reorganize Section 4. After talking about preprocessing, please start with unsupervised methods and then introduce supervised methods. Under each of these two, different sensors can be introduced separately.

17) Furthermore, in the Tables (7,8, ...) no need to introduce area or performance. Most of them are evaluated on quite different datasets. Such numbers will only
mislead readers. Just the first 4 coulmns are sufficient.

18) Table 7 (CD on SAR datasets), following works need to be added:
i) Reference 148
ii) Unsupervised change-detection based on convolutional-autoencoder feature extraction
iii) LCS-EnsemNet: A Semisupervised Deep Neural Network for SAR Image Change Detection With Dual Feature Extraction and Label-Consistent Self-Ensemble
iv) Dynamic Graph-Level Neural Network for SAR Image Change Detection
v) Change detection in image time-series using unsupervised lstm

19) Line 586: "change trajectory maps" - change to "change detection maps"

20) Table 8: some works, e.g., citation 128 needs to be be excluded. This work was published in a journal of low impact factor and has not been cited in 4 years
after publication. Why do the authors think that this is a mentionworthy work here?

21) Table 8: following works need to be discussed:
i) Unsupervised Deep Change Vector Analysis for Multiple-Change Detection in VHR Images
ii) A Feature Difference Convolutional Neural Network-Based Change Detection Method
iii) Unsupervised deep transfer learning-based change detection for hr multispectral images
iv) Sentinel-2 change detection based on deep features
v) Semisupervised change detection using graph convolutional network
vi) A Deep Multitask Learning Framework Coupling Semantic Segmentation and Fully Convolutional LSTM Networks for Urban Change Detection
vii) A multiscale graph convolutional network for change detection in homogeneous and heterogeneous remote sensing images
viii)  Unsupervised deep joint segmentation of multitemporal high-resolution images

22) Furthermore, one interesting direction in supervised optical change detection is trustworthiness of models, as discussed in "Trusting small training dataset for supervised change detection". This needs to be discussed.

23) Table 9, following works need to be discussed:
i) Empowering change vector analysis with autoencoding in bi-temporal hyperspectral
images
ii) Change Detection in Hyperdimensional Images Using Untrained Models

24) There are some discrepancies in page 22 and Page 23. Please note line 822 reference 148 (author name) and reference 148 in Table 10! Similarly, please note, 
reference 149 in Table 10 and the corresponding entry in line 1501 page 36. They do not match!

25) Some works in comment 21 also needs to e added here in Table 10.

26) Line 859: I would not call 5 m a VHR image.

27) Table 11: following works need to be discussed:
i) Unsupervised multiple-change detection in VHR multisensor images via deep-learning based adaptation
ii) Sentinel-1 and Sentinel-2 Data Fusion for Urban Change Detection using a Dual Stream U-Net
iii) Self-supervised Multisensor Change Detection
iv) Selective Adversarial Adaptation-Based Cross-Scene Change Detection Framework in Remote Sensing Images
v) Fusing multi-modal data for supervised change detection
vi) A deep siamese network with hybrid convolutional feature extraction module for change detection based on multi-sensor remote sensing images
vii) Mitigating spatial and spectral differences for change detection using super-resolution and unsupervised learning

28) In addition to the described sensors, one emerging topic is planetary change detection. Please add a brief discussion on this. Following works
can be discussed:
i) Toward Generalized Change Detection on Planetary Surfaces With Convolutional Autoencoders and Transfer Learning
ii) Patch-level unsupervised planetary change detection

29) Section 5: Evaluation techniques: many CD works use sensitivity (accuracy of changed pixels) and specificity (accuracy of unchanged pixels). Please discuss them

30) Discussion, especially on multiple change map is slightly lacking. Among the works that has been mentioned in the points above, some of them also allows to obtain
multiple CD map, e.g., 21.i or 23.ii or 27.i. Please discuss them briefly here.

31) "Still, it has been seen that deep learning is not interpreted so quickly, and it requires extensive datasets as it does not
perform outstanding results on smaller datasets." - This is only partially true in change detection where an extensive set of work has been
done on the unsupervised methods. What is lacking in this manuscript currently is due consideration to the unsupervised methods, which is leading to a partially wrong conclusion. Addressing previous points will address this issue and the conclusion also needs to be revised accordingly.

Author Response

Thank you for investing your time and effort in our manuscript. We have seriously considered all of your comments and suggestions and improved the manuscript accordingly.

Reviewer 2 Report

Dear Authors,

The main objective of this paper is about DL-based CD in RS images: a review.

Specific comments:

  1. This paper is a kind of Review paper, not an article. The authors must change “Article” to “Review” at the line above the title of the manuscript.
  2. The main objective of the title of the paper is investigating the DL-based CD in RS images but except for some tables (.7, to .10), which list a few names of DL methods, the entire paper is about data and various sensors used in detecting changes. It is necessary to go deeper into the concepts of DL and its application in detecting change.
  3. Current approach of novel change detection methods is proposing the “multiple changes” beside “binary changes”, but unfortunately, this paper does not say anything about it except ref [176].
  4. It is suggested that the authors announce the download link of the CD images in the part of the paper for the readers.
  5. Novelty or contribution: the authors should summarize clearly what the novelty or contribution of this work is, or what is the difference/superiority of this work compared with the existing methods, especially in the abstract and introduction section?

Author Response

Thank you for investing your time and effort in our manuscript. We have seriously considered all of your comments and suggestions and improved the manuscript accordingly

Author Response

(The authors gave the same response as above.)

Reviewer 4 Report

The paper presents a survey on remote sensing change detection. Comments are listed as follows:

1) Motivation and main contributions are not clear to attract readers. Some other works (e.g., https://ieeexplore.ieee.org/abstract/document/9136674, https://www.mdpi.com/2072-4292/12/10/1688/htm) have already been published for the given task.

2) The author requires adding a background section related to remote sensing change detection.

3) A meta-analysis process for data extraction needs to be added to the manuscript.

4) It would be good to see the process of extracting relevant papers based on the diverse combination of essential search expressions in a general flowchart.

5) It will be useful to present generic architectures relevant to the types of deep learning methods applied for change detection from different datasets.

6) It is interesting to add the advantages and disadvantages of the models to the tables for the summary of literature work of change detection techniques based on different datasets.

7) In the discussion section of this review, I expected to see a large discussion of the benefit of using this for various use cases. It would be good to put some results obtained with different methods on different cases in this part and discuss.

8) It is also good to compare the quantitative results for some evaluation metrics based on a common reference base achieved by multiple DL methods for change detection from various datasets by investigating presentation graphs.

9) The conclusion is shallow and should highlight the main outcomes of the study in detail.

10) The following papers are also good to be reviewed and improve the paper.

[1] https://doi.org/10.1109/ACCESS.2020.3008036

[2] https://doi.org/10.3390/rs13183710

Author Response

(The authors gave the same response as above.)

Round 2

Reviewer 1 Report

The manuscript is vastly improved from its previous version and I am satisfied with the revised version. Only few minor comments remain:

1) English can be increased at few places, e.g., just couple of examples from Page 1 (similarly such problems remain for the other pages, too):
i) Line 10: such as SAR,Multispectral,Hyperspectral - I would have expected space after all commas
ii) Line 27: "exposed new academic challenges" - though not an error, however sounds strange. Needs to be revised.

2)Table 1: citation, please mention the citation source.

3) Line 163: It is not clear why resolution is important property of SAR? SAR images can be of varying resolutions, just like other remote sensing sensors.

4) Line 164: MSRS or MS RS?

5) Table 2: please recheck Sentinel-5 resolution

6) Table 13: Prex et. al. -> Prexl et. al.

7) Page 29: Equation 2: please insert a space between sensitivity and or. Alternatively, you can use /

8) Future Scope: I would have expected a bit more discussion here.

Author Response

(The authors gave the same response as above.)

Reviewer 2 Report

All ambiguity is resolved.

Author Response

Thank you 

Reviewer 3 Report

Thanks authors addressing my comments. 

Author Response

Thank you

Reviewer 4 Report

The quality of the paper has significantly improved. However, my previous comments have not been fully addressed and I expect the authors to address the comments in this round carefully.

First, I asked to add the advantages and disadvantages of the models (some key points for rapid assessment) to the tables for summary of literature work of change detection techniques based on different datasets.

Second, I asked to compare the quantitative results for some evaluation metrics based on a common reference base achieved by multiple DL methods for change detection from various datasets by investigating presentation graphs.

Third, I commented that the conclusion is shallow and should highlight the main outcomes of the study in detail, however, I did not see any improvement in the new version.

Last, the following papers are also good to be reviewed and improve the paper.

[1] https://doi.org/10.1109/ACCESS.2020.3008036

[2] https://doi.org/10.3390/rs13183710

Author Response

Thank you for investing your time and effort in our manuscript. We have seriously  considered all of your comments and suggestions and improved the manuscript accordingly. Below, we answer all of your concerns, and we highlighted them in the manuscript in “orange” color.
